# Mannose receptor is an HIV restriction factor counteracted by Vpr in macrophages

Jay Lubow[1], Maria C Virgilio[2], Madeline Merlino[3], David R Collins[1], Michael Mashiba[4], Brian G Peterson[5], Zana Lukic[3], Mark M Painter[4], Francisco Gomez-Rivera[4], Valeri Terry[3], Gretchen Zimmerman[4], Kathleen L Collins[2,3,4]*

[1]Department of Microbiology and Immunology, University of Michigan, Ann Arbor, United States; [2]Cellular and Molecular Biology Program, University of Michigan, Ann Arbor, United States; [3]Department of Internal Medicine, University of Michigan, Ann Arbor, United States; [4]Graduate Program in Immunology, University of Michigan, Ann Arbor, United States; [5]Department of Biological Chemistry, University of Michigan, Ann Arbor, United States

**Abstract** HIV-1 Vpr is necessary for maximal HIV infection and spread in macrophages. Evolutionary conservation of Vpr suggests an important yet poorly understood role for macrophages in HIV pathogenesis. Vpr counteracts a previously unknown macrophage-specific restriction factor that targets and reduces the expression of HIV Env. Here, we report that the macrophage mannose receptor (MR), is a restriction factor targeting Env in primary human monocyte-derived macrophages. Vpr acts synergistically with HIV Nef to target distinct stages of the MR biosynthetic pathway and dramatically reduce MR expression. Silencing MR or deleting mannose residues on Env rescues Env expression in HIV-1-infected macrophages lacking Vpr. However, we also show that disrupting interactions between Env and MR reduces initial infection of macrophages by cell-free virus. Together these results reveal a Vpr-Nef-Env axis that hijacks a host mannose-MR response system to facilitate infection while evading MR's normal role, which is to trap and destroy mannose-expressing pathogens.

*For correspondence: klcollin@umich.edu

Competing interests: The authors declare that no competing interests exist.

## Introduction

Vpr is a highly conserved HIV accessory protein that is necessary for optimal replication in macrophages (*Balliet et al., 1994*) but its mechanism of action is poorly understood. Studies using human lymphoid tissue (HLT), which are rich in both T cells and macrophages, have found that loss of Vpr decreases virus production (*Rucker et al., 2004*) but only when the virus strain used is capable of efficiently infecting macrophages (*Eckstein et al., 2001*). These studies provide evidence that Vpr enhances infection of macrophages and increases viral burden in tissues where macrophages reside. Because Vpr is packaged into the virion (*Cohen et al., 1990*) and localizes to the nucleus (*Lu et al., 1993*), it may enhance early viral replication events. However, in mononuclear phagocytes *vpr*-null virus in which Vpr protein is provided by trans-complementation in the producer cells replicates poorly compared to wild-type virus (*Connor et al., 1995*), indicating that Vpr's role in the HIV replication cycle continues into late stages.

Previous work by our group demonstrated that Vpr counteracts an unidentified macrophage-specific restriction factor that targets Env and Env-containing virions for lysosomal degradation (*Mashiba et al., 2014*; *Collins et al., 2015*). This restriction could be conferred to permissive 293T cells by fusing them with MDM to create 293T-MDM heterokaryons. A follow up study demonstrated

**eLife digest** Human cells have defense mechanisms against viral infection known as restriction factors. These are proteins that break down parts of a virus including its DNA or proteins. To evade these defenses, viruses in turn make proteins that block or break down restriction factors. This battle between human and viral proteins determines which types of cells are infected and how quickly a virus can multiply and spread to new cells.

HIV produces a protein called Vpr that counteracts a restriction factor found in immune cells called macrophages. However, the identity of the restriction factor targeted by Vpr is a mystery. When Vpr is missing, this unknown restriction factor breaks down a virus protein called Env. Env is a glycoprotein, which is a protein with sugars attached. When Env levels are low, HIV cannot spread to other cells and multiply. Identifying the restriction factor that breaks down Env may lead to new ways of treating and preventing HIV infections.

Now, Lubow et al. reveal that the unknown restriction factor in macrophages is a protein called the mannose receptor. This protein binds and destroys proteins containing mannose, a type of sugar found on bacteria and some viruses. The experiments revealed that the mannose receptor grabs mannose on the HIV protein Env. This causes Env to be broken down and stops HIV from spreading.

Lubow et al. also find that Vpr works with another protein produced by HIV called Nef to reduce the number of mannose receptors on macrophages. The two proteins do this by targeting different steps in the assembly of mannose receptors, allowing the virus to multiply and spread more efficiently. The experiments suggest that drugs that simultaneously block Vpr and Nef might prevent or suppress HIV infections. More studies are needed to develop and test potential HIV-treatments targeting Vpr and Nef.

that by increasing steady state levels of Env, Vpr increases formation of virological synapses between infected MDM and autologous uninfected T cells, enhancing HIV infection of T cells (*Collins et al., 2015*). This enhances spread to T cells and dramatically increases levels of Gag p24 in the culture supernatant. This finding helps explain the paradoxical observations that Vpr is required for maximal infection of T cells in vivo (*Hoch et al., 1995*) but numerous studies have shown Vpr only marginally impacts infection of pure T cell cultures in vitro (e.g. *Mashiba et al., 2014*).

Our goal in the current study was to identify and characterize the myeloid restriction factor targeting Env that is counteracted by Vpr. We reasoned that macrophage-specific Env-binding proteins, including the carbohydrate binding protein mannose receptor (MR), were candidates. MR is expressed on several types of macrophages in vivo (*Liang et al., 1996*; *Linehan et al., 1999*) and is known to mediate innate immunity against various pathogens (*Macedo-Ramos et al., 2014*; *Subramanian et al., 2019*). MR recognizes mannose-rich structures including high-mannose glycans, which are incorporated in many proteins during synthesis. In eukaryotic cells most high-mannose glycans are cleaved by α-mannosidases and replaced with complex-type glycans as they transit through the secretory pathway. By contrast, in prokaryotic cells, high-mannose residues remain intact, making them a useful target of pattern recognition receptors including MR. Some viral proteins, including HIV-1 Env, evade mannose trimming (*Coss et al., 2016*) and retain enough high-mannose to bind MR (*Trujillo et al., 2007*; *Lai et al., 2009*). There is evidence that HIV-1 proteins Nef and Tat decrease expression of MR based on studies performed in monocyte derived macrophages (MDM) and the monocytic U937 cell line, respectively (*Caldwell et al., 2000*; *Vigerust et al., 2005*). Nef dysregulates MR trafficking using an SDXXLΦ motif in MR's cytoplasmic tail (*Vigerust et al., 2005*), which is similar to the sequence in CD4's tail that Nef uses to remove it from the cell surface (*Bresnahan et al., 1998*; *Greenberg et al., 1998*; *Cluet et al., 2005*). Whether MR or its modulation by viral proteins alters the course of viral replication has not been established.

Here we confirm that Nef reduces MR expression in primary human MDM, although in our system, the effect of Nef alone was relatively small. In contrast, we report that co-expression of Vpr and Nef dramatically reduced MR expression. In the absence of both Vpr and Nef, MR levels normalized indicating that Tat did not play a significant, independent role in MR downmodulation. Deleting mannose residues on Env or silencing MR alleviated mannose-dependent interactions between MR and Env and reduced the requirement for Vpr. Although the post-infection interactions between MR

and Env reduced Env levels and inhibited viral release, we provide evidence that these same interactions were beneficial for initial infection of MDM. Together these results reveal that mannose residues on Env and the accessory proteins Nef and Vpr are needed for HIV to utilize and then disable an important component of the myeloid innate response against pathogens intended to thwart infection.

## Results

### Identification of a restriction factor counteracted by Vpr in primary human monocyte-derived macrophages

Because we had previously determined that Vpr functions in macrophages to counteract a macrophage specific restriction factor that targets Env, we reasoned that Env-binding proteins selectively expressed by macrophages were potential candidate restriction factors. To determine whether any factors fitting this description were targeted by Vpr, we cultured macrophages under conditions that achieve a saturating infection by both wild-type and Vpr-null mutant viruses (*Figure 1A and B*). We found that mannose receptor (MR), which is highly expressed on macrophages and has been previously shown to bind Env (*Trujillo et al., 2007*; *Fanibunda et al., 2008*; *Lai et al., 2009*), was significantly decreased by wild-type HIV 89.6 but not by 89.6 *vpr*-null (*Figure 1C and D*, p<0.01). In contrast, we observed no significant effect of Vpr on the expression of GAPDH. We also observed that stimulator of interferon genes (STING) was unaffected by Vpr (*Figure 1—figure supplement 1*). Relative expression of known restriction factors GBP5 and IFITM3 varied in infected MDM from multiple donors (*Figure 1—figure supplement 1*), but unlike MR they were not consistently reduced in the wild-type condition, indicating they are not targeted by Vpr.

To confirm the effect of Vpr on Env during HIV infection of primary human macrophages in which MR was downmodulated, we performed quantitative western blot analysis. As shown in *Figure 1E and F*, we confirmed that amounts of Vpr sufficient for MR downmodulation were also sufficient for stabilizing expression of Env (gp160, gp120, gp41). Compiled data from nine donors clearly demonstrated results that were similar to our prior publication (*Mashiba et al., 2014*); under conditions of matched infection in which there was no significant difference in HIV Gag pr55 levels between wild-type and *vpr*-null infections, all three forms of Env were significantly more abundant in the wild-type infection (gp160: 4-fold, p<0.002; gp120: 6-fold, p<0.002; gp41: 3-fold, p<0.001).

### Vpr and Nef counteract MR expression in infected macrophages via independent and additive mechanisms

Because an earlier report indicated that Nef decreases surface expression of MR (*Vigerust et al., 2005*), we asked whether Nef was playing a role in MR downmodulation in our systems. Because HIVs lacking Vpr and Nef spread too inefficiently in MDM to observe effects on host proteins by western blot analysis, we utilized a replication defective HIV with a GFP marker (NL4-3 ΔGPE-GFP, *Figure 2A*) to allow measurement of MR expression via flow cytometry following single-round transduction. This construct has the additional advantage that it eliminates potentially confounding effects of differences between wild-type and mutant HIV viral spread. We generated truncation mutations in *nef* and *vpr* and confirmed that these mutations only affected expression of the altered gene product in transfected 293T (*Figure 2B*). For these experiments, primary MDM were harvested earlier than the experiments described in *Figure 1* (five days versus ten days) because the viruses could not replicate and the GFP marker allowed identification of transduced cells (*Figure 2C*). Under these conditions, we found that MR expression was dramatically reduced in a subset of GFP⁺ cells when both Vpr and Nef were expressed (*Figure 2C–E*). Both Nef and Vpr contributed to MR downmodulation; loss of function mutation in either Vpr or Nef reduced the severity of MR downmodulation similarly, and there was no statistical difference between MR levels in macrophages expressing either Vpr or Nef alone (*Figure 2E*). In addition, complete elimination of downmodulation required mutation of both Vpr and Nef (*Figure 2C–E*). These results indicate that both Vpr and Nef are required for maximal MR downmodulation in HIV-infected macrophages and that neither alone is sufficient.

Vpr was previously demonstrated to interact with a cellular co-factor called DCAF1, a component of the cellular DCAF1-DDB1-CUL4 E3 ubiquitin ligase complex. (*Hrecka et al., 2007*; *Le Rouzic*

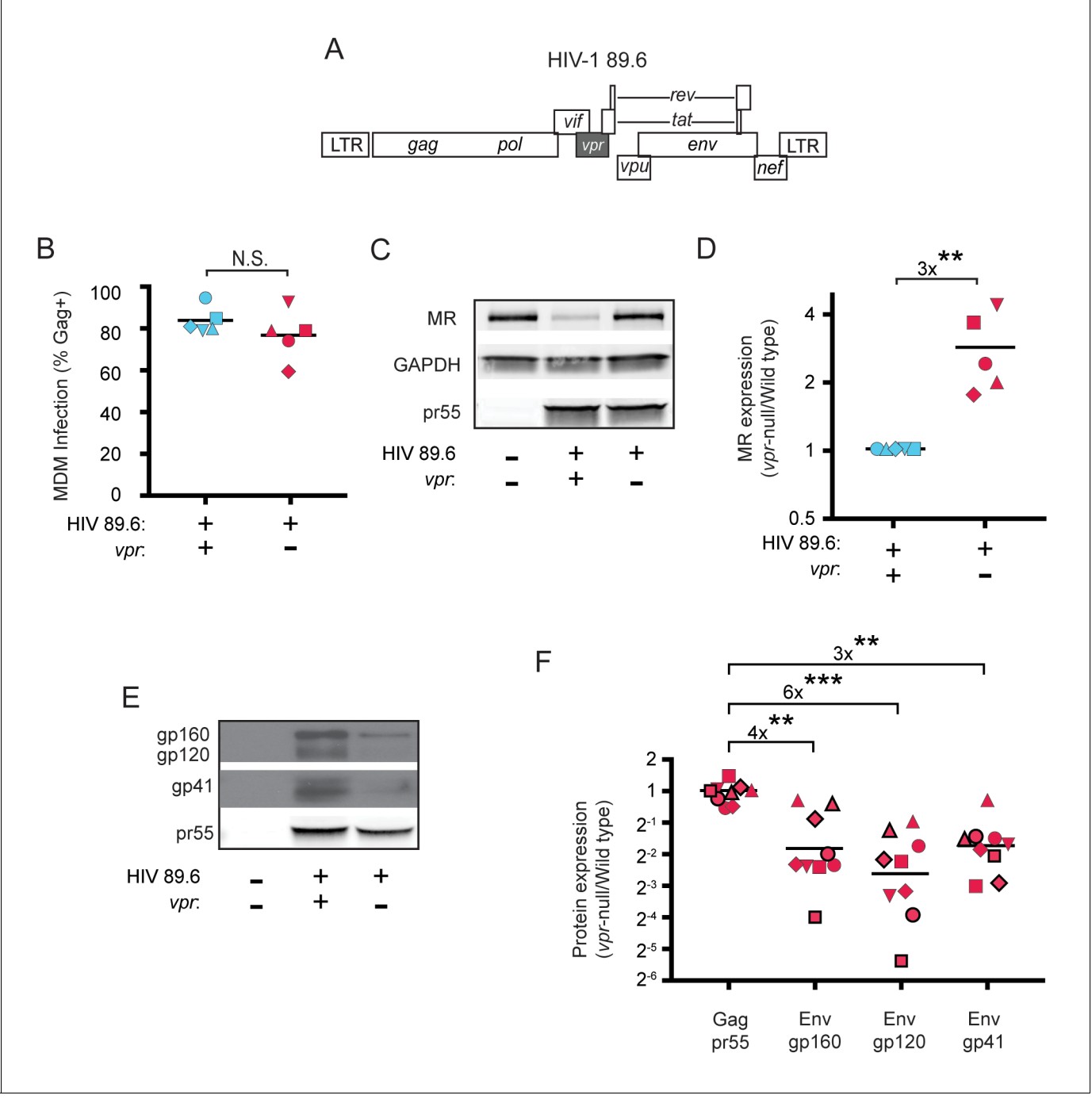

**Figure 1.** HIV Vpr reduces steady state levels of host mannose receptor in MDM and increases steady state levels of viral Env protein. (A) Diagram of the HIV 89.6 proviral genome. The shaded box shows the location of *vpr*, which was disrupted by a frame shift mutation to create the Vpr-null version (*Mashiba et al., 2014*). HIV-1 89.6 is a dual CXCR4/CCR5-tropic HIV molecular clone isolated from the peripheral blood of an AIDS patient (*Collman et al., 1992*). (B) Summary graph depicting MDM infected by HIV 89.6 wild-type and *vpr*-null with matched infection frequencies of at least 50% 10 days post infection as measured flow cytometrically by intracellular Gag p24 staining. This subset with high frequencies of infection was selected to examine potential effects on host factors. (C) Western blot analysis of whole cell lysates from MDM prepared as in B. (D) Summary graph displaying relative expression of MR in wild-type and mutant 89.6 from blots as shown in C. Western blot protein bands were quantified using a Typhoon scanner. Values for MR expression in MDM infected with Vpr-null HIV were normalized to GAPDH and then to wild-type for each donor. Statistical significance was determined using a two-tailed, ratio *t*-test. **p=0.005 (E) Western blot analysis of HIV protein expression in MDM infected as in B. (F) Summary graph of HIV protein expression from western blot analysis as in E and quantified as described in methods. The ratio of expression in wild-type to *vpr*-null infection is shown. Data from 9 independent donors with similar frequencies of infection (within 2-fold) following ten days of

*Figure 1 continued on next page*

*Figure 1 continued*

infection are shown. Statistical significance was determined using a two-tailed, ratio *t*-test, N.S. – not significant, p=0.31, **p<0.01, ***p<0.001. Data from each donor is represented by the same symbol in all charts. Mean values are indicated.

The online version of this article includes the following figure supplement(s) for figure 1:

**Figure supplement 1.** HIV Vpr reduces steady state levels of MR but not GBP5, STING or IFITM3.

---

*et al., 2007*; *McCall et al., 2008*; *Lahouassa et al., 2016*; *Wu et al., 2016*; *Zhou et al., 2016*). The interaction between Vpr and DCAF1 can be disrupted through a Vpr mutation (Q65R) that inhibits many Vpr-dependent functions, including reversal of Env degradation in macrophages (*Mashiba et al., 2014*). To determine whether this mutant is defective at MR downmodulation, we generated the mutation in the NL4-3 ΔGPE-GFP parent (*Figure 2A*), confirmed expression in transfected 293T cells (*Figure 2F*) and tested the effect of the mutation on MR levels in macrophages. As expected, we found that in transduced MDM the *vpr*-Q65R mutant behaves similarly to *vpr*-null (*Figure 2E*). These results indicate interactions between Vpr and DCAF1 are required to mediate Vpr's effects on MR.

The differences in MR downmodulation we observed using this system were not due to variations in multiplicity of infection of the different viral constructs as MDM transduced with the mutant viral constructs had roughly similar transduction rates as the parental construct (*Figure 2G*) but demonstrated less MR downmodulation (*Figure 2E*).

To determine whether the relatively modest effect of Nef alone on MR levels was due to using HIV to deliver Nef as compared to an adenoviral vector delivery system used in a prior publication (*Vigerust et al., 2005*), we repeated the experiment using an adenoviral vector expressing Nef. These experiments confirmed that levels of Nef sufficient to downmodulate the HIV receptor, CD4, on nearly all MDM in the culture achieved only modest effects on MR in a subset of cells (*Figure 2H*) similar to what was observed using the HIV reporter construct (*Figure 2E*). Thus, Nef and Vpr have modest but significant effects on MR when expressed individually, however the combined effects of both proteins can achieve nearly complete downmodulation at least in a subset of infected cells.

While the effect of Nef has been previously reported and found to be due to disruption of MR intracellular trafficking (*Vigerust et al., 2005*), the effect of Vpr on MR is a novel observation. Vpr is known to target cellular proteins involved in DNA repair pathways for proteasomal degradation via interactions with Vpr binding protein [DCAF1, (*McCall et al., 2008*)]. Using this mechanism, Vpr degrades the uracil deglycosylases UNG2 and SMUG1 in 293T cells following co-transfection (*Schröfelbauer et al., 2005*; *Schröfelbauer et al., 2007*). To determine whether Vpr directly targets MR using a similar strategy, we co-transfected NL4-3 ΔGPE-GFP or a *vpr*-null derivative with expression vectors encoding an UNG2-FLAG fusion protein or MR (*Liu et al., 2004*) in 293T cells. We then analyzed expression of MR or UNG2 by flow cytometry and western blot (*Figure 2—figure supplement 1*). We found that Vpr in 293T cells virtually eliminated UNG2 expression when measured by flow cytometry and noticeably reduced UNG2 by western blot. However, Vpr had no effect on expression of MR measured by either method. Thus, we concluded that Vpr does not degrade MR by the direct, proteasomal mechanism it uses to degrade UNG2. Because MR expression in this system is controlled by a heterologous CMV promoter; the lack of effect by Vpr suggested its action may depend on MR's native promoter.

## Vpr reduces transcription of MRC1

In addition to targeting proteins for degradation, Vpr also functions to inhibit transcription of genes such as *IFNA1* (*Laguette et al., 2014*; *Mashiba et al., 2014*). Therefore, we hypothesized that Vpr may reduce MR expression via inhibition of transcription. To examine this, we assessed transcriptional activity in primary human MDM transduced with the wild-type or Vpr-null reporter virus (*Figure 3A*) using cells isolated based on GFP expression (*Figure 3B*). We found that the MR gene (*MRC1*) was consistently reduced in cells transduced by *vpr*-competent virus compared to cells transduced by *vpr*-null virus (*Figure 3C and D*, p=0.001). In contrast, any effects of Vpr on the housekeeping genes *ACTB* (β-actin) and *POL2A* (RNA polymerase 2A) were significantly smaller (*Figure 3D*, p<0.01). Similar results were obtained when each gene was normalized to *ACTB* instead of *GAPDH* (*Figure 3—figure supplement 1A–B*). The magnitude of the effect on *MRC1* is consistent

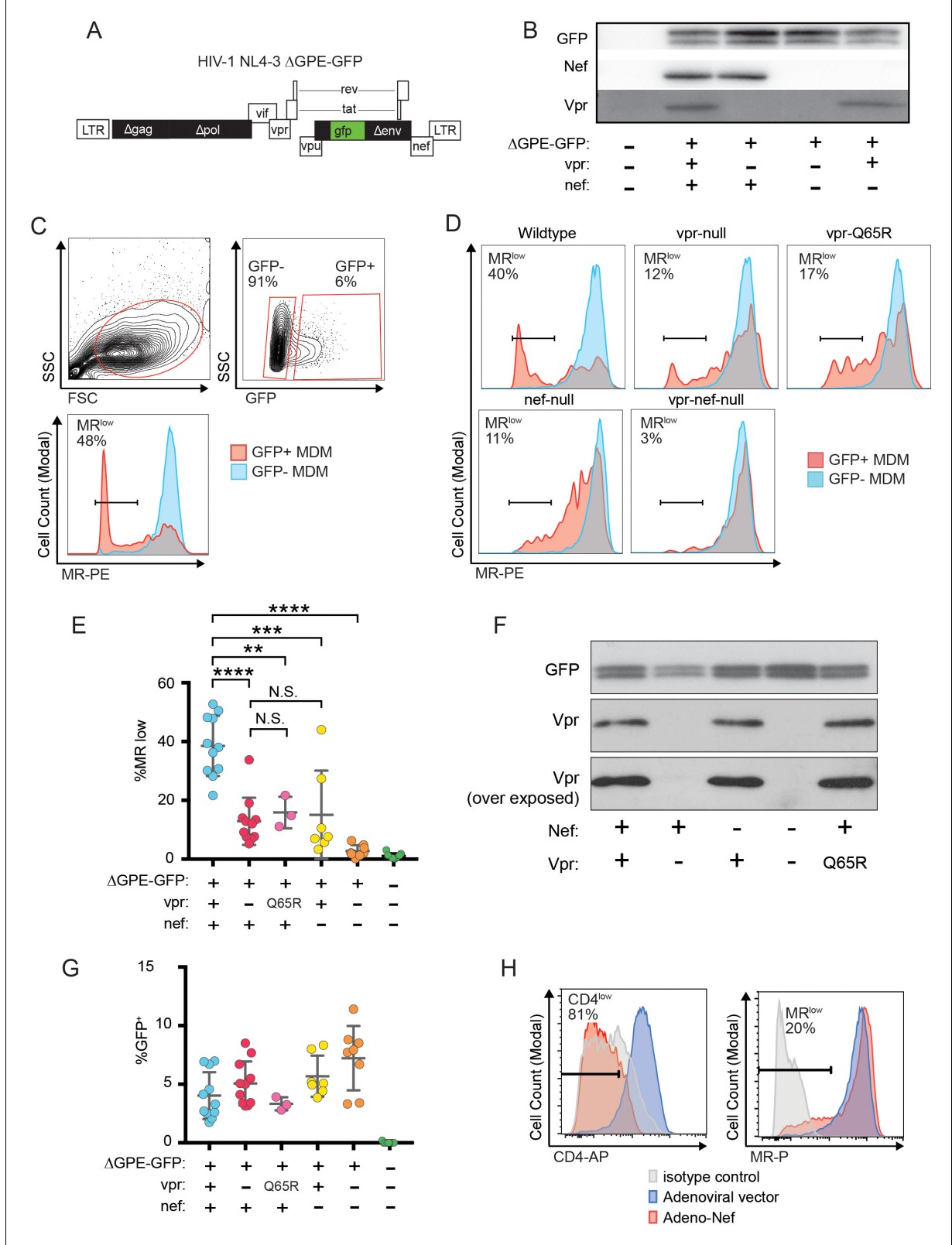

**Figure 2.** Combined effects of Nef and Vpr completely remove MR from a significant proportion of infected cells at early time points. (A) Diagram of HIV NL4-3 ΔGPE-GFP. (B) Western blot analysis of whole cell lysates from 293T cells transfected with the indicated viral expression construct. (C) Flow cytometry plots indicating the gating strategy used to identify live GFP+ vs GFP- cells and the fraction of cells that are MRlow. (D) Representative flow cytometric analysis of MDM at five days post transduction by the indicated virus. The percentage of GFP+ cells that fell into the MRlow gate is indicated

*Figure 2 continued on next page*

Figure 2 continued

in each panel. (E) Summary graph depicting the percentage of GFP$^+$ cells that fell into the MR$^{low}$ gate in transduced MDM. For the uninfected column the results from GFP$^-$ cells are displayed. (Each dot indicates an independent donor, range 3–11). (F) Western blot analysis of whole cell lysates from 293T cells transfected with the indicated viral expression construct. (G) Summary graph depicting the frequency of transduced (GFP$^+$) MDM at the time of harvest. (H) Representative flow cytometric plots of MDM transduced with the indicated adenoviral vector (n = 3 independent donors). For parts E and F mean +/- standard deviation is shown. Statistical significance was determined by a two-tailed, paired t-test. N.S. not significant, **p<0.01, ***p<0.001, ****p<0.0001.

The online version of this article includes the following figure supplement(s) for figure 2:

**Figure supplement 1.** HIV Vpr reduces steady state levels of UNG2 but not MR in co-transfected 293T cells.

with prior reports of HIV-1 inhibiting *MRC1* transcription— though this was not previously linked to Vpr (*Koziel et al., 1998*; *Sukegawa et al., 2018*). Relative *MRC1* expression in untransduced MDM was heterogeneous, varying over a ten-fold range. When compiled across donors, *MRC1* levels in mock-transduced samples were not significantly different than transduced (*Figure 3—figure supplement 1C–F*).

## Combined effect of Vpr and Nef dramatically enhances Env levels in primary human MDM

To determine whether the striking downmodulation of MR we observed with expression of both Nef and Vpr affected viral spread in MR$^+$ macrophages, we generated additional mutations in HIV-1 89.6 to create a *nef*-null mutant and a *vpr-nef*-null double mutant. As expected, in transfected 293T cells these mutations did not alter Env protein levels (*Figure 4A*) or release of virions as assessed by measuring Gag p24 in the supernatant by ELISA (*Figure 4B*). However, in primary human MDM infected with these HIVs, the mutants demonstrated defects in viral spread, with the double mutant having the greatest defect (*Figure 4C and D*). The defect in spread was caused in part by diminished virion release, which we previously showed occurred in the absence of Vpr (*Mashiba et al., 2014*); MDM infected with the HIV mutants released less Gag p24 even after adjusting for the frequency of infected cells (*Figure 4D*, right panel).

To determine whether combined effects of Nef and Vpr on MR expression affected Env restriction, we assessed Env levels in primary human MDM infected with each construct. Because the frequency of infected cells as assessed by intracellular Gag staining (*Figure 4C*) and Gag pr55 western blot (*Figure 4E*) was lower in the mutants than in the wild-type infection, lysate from the wild-type sample was serially diluted to facilitate comparisons. Remarkably, we found that the *vpr-nef*-null double mutant, which retains near normal MR levels, exhibited the greatest defect in Env expression (*Figure 4E*, compare lanes with similar Gag as indicated). In sum, Vpr and Nef-mediated downmodulation of MR correlated inversely with Env levels, consistent with MR being the previously described but unidentified HIV restriction factor that targets Env for lysosomal degradation in macrophages and is counteracted by Vpr (*Mashiba et al., 2014*). Combined effects of Nef on MR and other Env binding proteins including CD4 (*Aiken et al., 1994*) and chemokine receptors (*Michel et al., 2006*) may also play a role in stabilization of Env.

## Mannose-containing glycans in Env are required for macrophage restriction of HIV in the absence of Vpr

A particularly dense mannose-containing structure on Env, known as the mannose patch, may mediate interactions between Env and MR. This structure is present on all HIV Env proteins that require Vpr for stability in macrophages [89.6, NL4-3 and AD8 (*Mashiba et al., 2014*; *Collins et al., 2015*). Interestingly, a macrophage tropic strain YU-2, which was isolated from the CNS of an AIDS patient (*Li et al., 1991*), lacks a mannose patch. This structure is the target of several broadly neutralizing antibodies including 2G12, to which YU-2 is highly resistant (*Trkola et al., 1996*). If Vpr targets MR to counteract detrimental interactions between MR and mannose residues on Env, we hypothesized that HIV Envs lacking a mannose patch would have a reduced requirement for Vpr. To test this hypothesis, we first examined the extent to which virion release and Env expression were influenced by Vpr in primary human MDM infected with YU-2 or 89.6 HIVs. Consistent with our hypothesis, we observed no significant difference in Gag p24 release between wild-type and *vpr*-null YU-2 infection

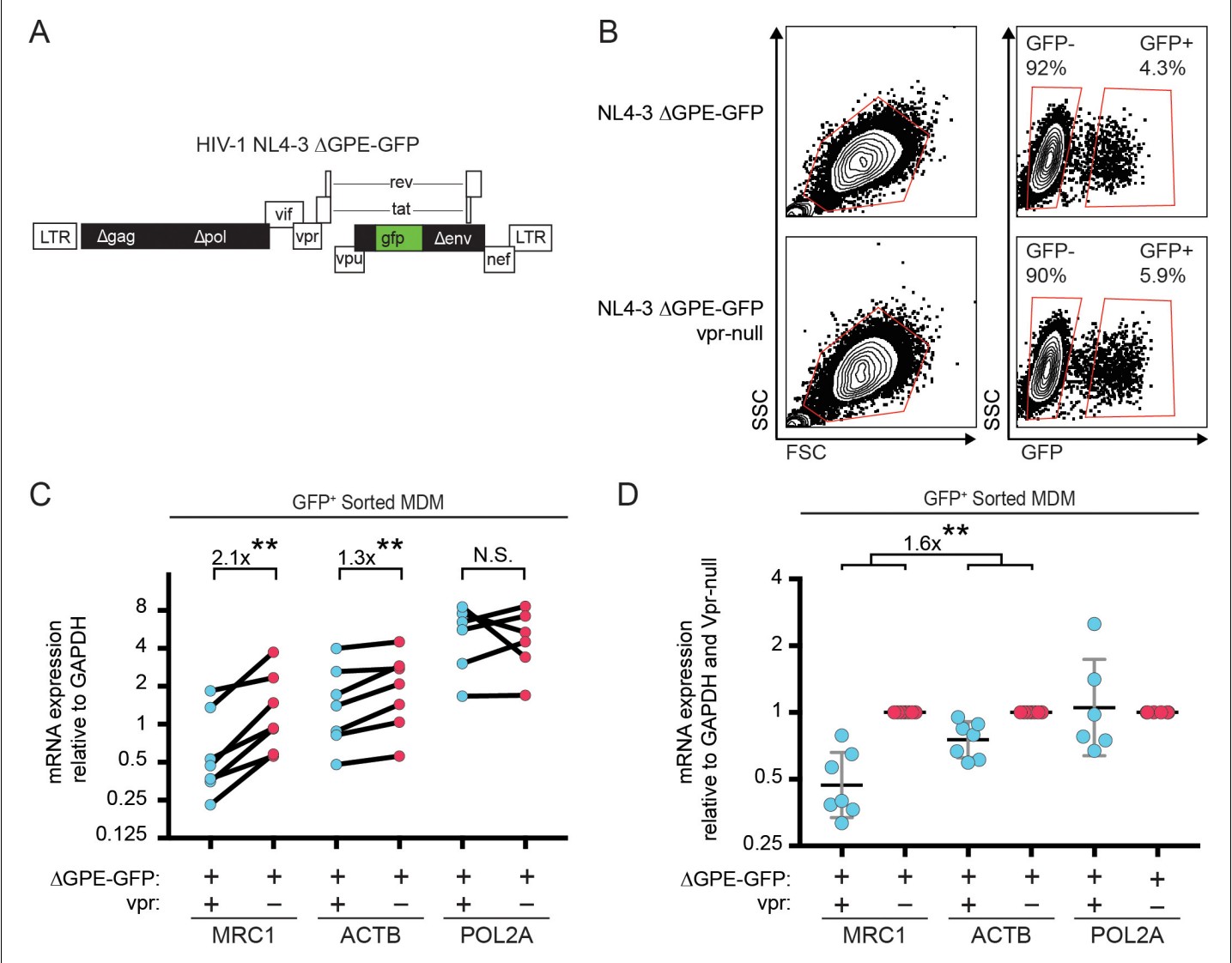

**Figure 3.** Vpr reduces transcription of *MRC1*. (**A**) Diagram of HIV NL4-3 ΔGPE-GFP. (**B**) Flow cytometry plots indicating the gating strategy used to sort live GFP⁺ vs GFP⁻ cells for subsequent qPCR analysis. (**C**) Summary graph of mannose receptor (*MRC1*), β-actin (*ACTB*) and RNA Polymerase 2A (*POL2A*) mRNA expression in MDM transduced with the indicated HIV reporter and sorted for GFP expression by FACS. All data are normalized to *GAPDH* mRNA expression. (**D**) Summary graph of *MRC1*, *ACTB* and *POL2A* expression normalized to the Vpr-null condition in each donor. (n = 7 independent donors). Geometric mean +/- geometric standard deviation is shown. Statistical significance was determined by a two-tailed, ratio *t*-test. N.S. = not significant p=0.81, **p<0.01.

The online version of this article includes the following figure supplement(s) for figure 3:

**Figure supplement 1.** Vpr reduces transcription of *MRC1*.

of MDM (*Figure 5A*). Moreover, the *vpr*-null mutant of YU2 displayed only a minor defect in Env expression compared to Vpr null versions of 89.6 and NL4-3 (*Figure 5B*).

Because there are a number of other genetic differences between YU-2 and the other HIVs, we constructed a chimeric virus, which restricted the differences to the *env* open reading frame. As shown in *Figure 5C*, a fragment of the YU-2 genome containing most of *env* but none of *vpr* (*Figure 5C*, shaded portion) was cloned into NL4-3 and NL4-3 *vpr*-null. As expected, these genetic alterations did not affect Env protein levels or virion release in transfected 293T cells (*Figure 5D and E*). To confirm that the chimeric Env was still functional, we examined infectivity in T cells prior to performing our analyses in primary human MDM. Conveniently, sequence variation within the gp120

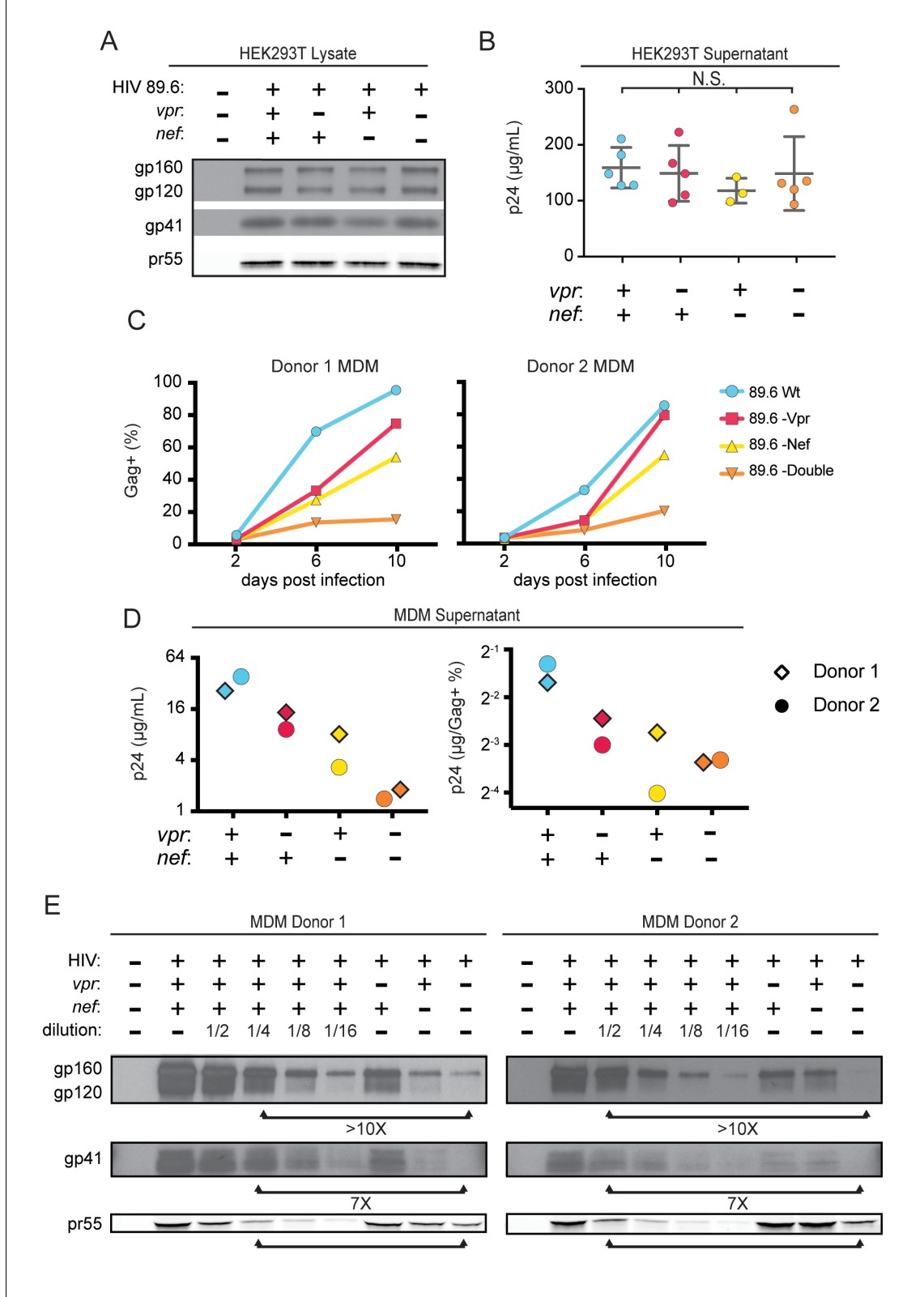

**Figure 4.** Combined effect of Vpr and Nef dramatically enhances Env levels in primary human MDM. (**A**) Western blot analysis of whole cell lysate from 293T transfected with the indicated HIV construct. (**B**) Summary graph of virion release from 293T cells transfected as in A and measured by Gag p24 ELISA. (*n* = 5 independent transfections). The mean +/- standard deviation is shown. Statistical significance was determined by one-way ANOVA. (N.S. – not significant) (**C**) Frequency of infected primary human MDM infected with the indicated HIV and analyzed over time by flow cytometric analysis of

*Figure 4 continued on next page*

*Figure 4 continued*

intracellular Gag. (For parts C-E, *n* = 2 independent donors). (**D**) Virion release by primary human MDM infected with the indicated HIV and analyzed by Gag p24 ELISA 10 days post infection. In the right panel, virion release was adjusted for frequency of infected cells as measured in part C. (**E**) Western blot analysis of whole cell lysate from primary human MDM infected with the indicated HIV. Within each donor, lanes 2–6 are a serial dilution series of the wild-type sample. The arrows below the Gag pr55 bands indicate the dilution of wild-type that has approximately the same amount of Gag pr55 as the *vpr-nef*-null double mutant.

region allows YU-2 Env to only utilize the co-receptor CCR5 for entry, whereas NL4-3 can only utilize CXCR4. Thus, we expected the NL4 3$env^{YU2}$ chimera would switch from being CXCR4- to CCR5-tropic. To test this, we utilized a T cell line expressing both chemokine receptors (MOLT4-R5) and selectively blocked entry via CXCR4 and CCR5 entry inhibitors [AMD3100 and maraviroc, respectively (*Figure 5F*)]. As expected, entry into MOLT4-R5 cells by NL4-3 was blocked by AMD3100 but not maraviroc, indicating CXCR4-tropism. The chimeric NL4-3 $env^{YU2}$ and wild-type YU-2 demonstrated the inverse pattern, indicating CCR5-tropism. These results demonstrated that we had made the expected changes in the chimeric Env without disrupting its capacity to infect cells.

To determine whether swapping a limited portion of YU-2 *env* into NL4-3 alleviated the requirement for Vpr, we examined Env expression and virion release in primary human MDM infected with these viruses. Because the parental NL4-3 virus required pseudotyping with a macrophage-tropic Env for entry and was unable to spread in MDM, all infections were treated with entry inhibitors AMD3100 and maraviroc starting at 48 hr after inoculation and maintained throughout the culture period to block subsequent rounds of infection. Consistent with our hypothesis that YU-2 Env lacked determinants necessary for the restriction that was alleviated by Vpr, we observed that wild-type NL4-3 Env but not chimeric NL4-3 $env^{YU2}$ required Vpr for maximal expression (*Figure 5G*). Moreover, MDM infected with the chimeric HIV had a reduced requirement for Vpr for maximal virion release (*Figure 5H* and *Figure 5—figure supplement 1*). This experiment provides strong evidence that the requirement for Vpr can be alleviated by genetic changes within the *env* open reading frame. These results are consistent with a model in which YU-2 *env* confers resistance to the effects of MR due to the absence of the mannose-rich structure on the YU-2 Env glycoprotein.

## Deletion of N-linked glycosylation sites in Env reduces Env restriction in HIV infected human primary MDM and diminishes the need for Vpr and Nef

To more directly assess the role of mannose in restricting expression of Env in HIV-1 infected primary human MDM, we engineered a version of 89.6 Env in which two N-linked glycosylation sites, N230 and N339 (HIV HxB2 numbering) were deleted by substituting non-glycosylated amino acids found at analogous positions in YU-2 Env (*Figure 6A*). The glycosylation sites N230 and N339 were selected because they contain high-mannose glycan structures (*Leonard et al., 1990*) that are absent in YU-2 Env. Loss of N230 limits neutralization by glycan specific antibodies (*Huang et al., 2014*). Loss of N339 decreases the amount of oligomannose ($Man_9GlcNAc_2$) present on gp120 by over 25%, presumably by opening up the mannose patch to processing by α-mannosidases (*Pritchard et al., 2015*). These substitutions (N230D and N339E) in 89.6 did not alter virion production (*Figure 6B*) or Env protein expression (*Figure 6C*) in transfected 293T cells.

To confirm that mutation of N230 and N339 disrupted the mannose patch on Env, we assayed the ability of 2G12, which recognizes epitopes in the mannose patch (*Sanders et al., 2002*; *Scanlan et al., 2002*) to neutralize wild-type and mutant Env. As shown in *Figure 6D*, wild-type but not mannose deficient N230D N339E Env was neutralized by 2G12. In addition, we found that these substitutions did not disrupt infection of a T cell line that does not express MR (*Figure 6E*). However, somewhat unexpectedly, we found that HIV containing the N230D N339E Env substitutions was approximately 40% less infectious to primary human macrophages expressing MR than the wild-type parental virus (*Figure 6E*, p=0.002). This macrophage-specific difference in infectivity suggested that mannose on Env may facilitate initial infection through interactions with MR, which is highly expressed on differentiated macrophages. To examine this possibility further, we asked whether soluble mannan, which competitively inhibits MR interactions with mannose containing glycans (*Shibata et al., 1997*), was inhibitory to HIV infection of macrophages. As a negative control, we tested 89.6 Δenv pseudotyped with vesicular stomatitis virus G-protein Env (VSV-G) which has only

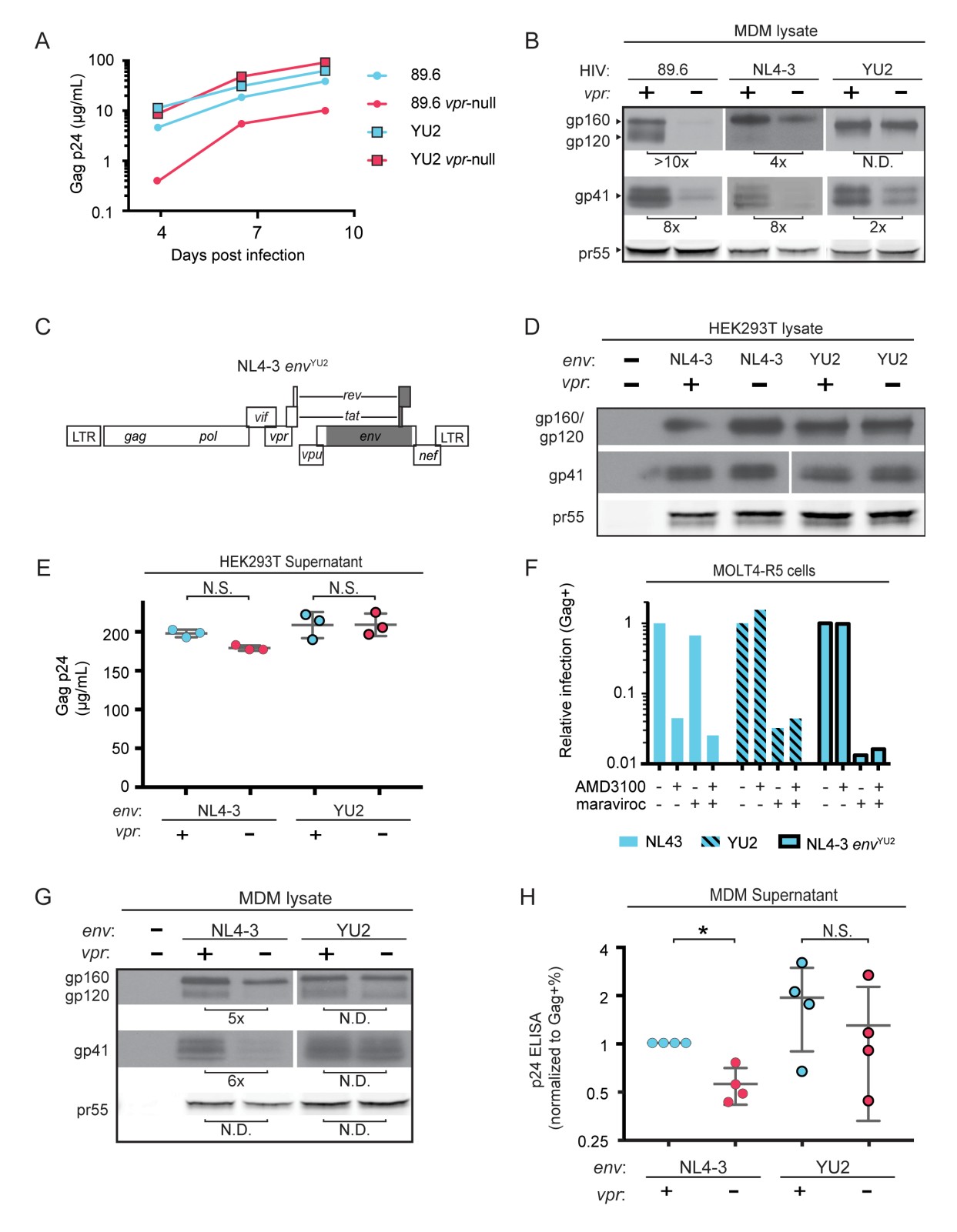

**Figure 5.** HIV YU2, which lacks a mannose-rich patch, does not require Vpr for robust Env protein expression and spread in MDM. (**A**) Virion release over time by primary human MDM infected with the indicated HIV as measured by ELISA (*n* = 2 independent donors). (**B**) Western blot analysis of whole cell lysates from MDM infected for 10 days with the indicated HIV. Because NL4-3 infects MDM poorly, NL4-3 was pseudotyped with a YU-2 Env expression plasmid co-transfected in the producer cells as described in Methods. Subsequent spread was blocked in all samples by the addition of

*Figure 5 continued on next page*

*Figure 5 continued*

entry inhibitors AMD3100 and maraviroc initially added 48 hr post-infection and maintained throughout the culture period. (C) Diagram of the HIV NL4-3 genome. The shaded portion represents the sequence that was replaced with sequence from HIV YU2 to create the NL4-3 env$^{YU-2}$ chimera. (D) Western blot analysis of 293T cells transfected with the indicated HIV constructs. YU-2 gp41 is detected by the monoclonal antibody z13e1 and NL4-3 gp41 is detected by the monoclonal antibody CHESSIE-8. (E) Virion release from 293T transfected as in D as measured by p24 ELISA. (n = 3 experimental replicates). (F) Relative infection of MOLT4-R5 cells 48 hr after inoculation with the indicated viruses and treatment with entry inhibitors as indicated. The frequency of infected cells was measured by intracellular Gag stain and normalized to the untreated condition for each infection. (G) Western blot analysis of primary human MDM infected for 10 days with the indicated virus as in B. (n = 2 independent donors). (H) Summary graph showing virion release as measured by p24 ELISA from primary human MDM infected as in G. Virus production was adjusted for infection frequency as determined flow cytometrically using an intracellular Gag stain. The mean +/- standard deviation is shown. (n = 4 independent donors). N.D. – no difference. Statistical significance was determined using a two-tailed, ratio *t*-test. N.S. – not significant, *p<0.05.

The online version of this article includes the following figure supplement(s) for figure 5:

**Figure supplement 1.** Raw p24 ELISA and intracellular Gag stain data following infection by NL4-3 env$^{YU2}$.

two N-linked glycosylation sites, both of which contain complex-type rather than high-mannose glycans (*Reading et al., 1978*). Therefore VSV-G should not bind MR or be inhibited by mannan. As expected, we found that infection of a T cell line lacking MR was not sensitive to mannan (*Figure 6F*, left panel). However, infection of MDM by wild-type HIV-1 was inhibited up to 16-fold by mannan (*Figure 6F*, right panel). This was specific to HIV Env because mannan did not inhibit infection by HIV lacking *env* and pseudotyped with heterologous VSV-G Env. Interestingly, mannan also inhibited baseline macrophage infection by mannose-deficient Env (89.6 Env N230D N339E), indicating that N230D N339E substitutions did not completely abrogate glycans on Env that are beneficial to initial infection. In sum, our results demonstrate that interactions with mannose binding receptors are advantageous for initial HIV infection of macrophages and that the glycans remaining on Env N230D N339E retain some ability to bind glycan receptors on macrophages that facilitate infection.

While interactions between high-mannose residues on Env and MR were advantageous for viral entry, we hypothesized that they interfered with intracellular Env trafficking and were deleterious to egress of Env-containing virions in the absence of Vpr and/or Nef. To test this, we examined virion release and Env expression by HIVs encoding the mannose-deficient Env N230D N339E in the presence or absence of Vpr. In a spreading infection of MDM, we found that virus expressing mannose-deficient Env had a reduced requirement for Vpr for maximal virus release compared with the parental wild-type virus (*Figure 6G*, p<0.001). In addition, in single-round infections of MDM, the mannose-deficient Env had a reduced requirement for both Nef and Vpr (*Figure 6H* and *Figure 6— figure supplement 1*, p<0.001). Single round infection assays cultured for ten days were used to assess the *vpr-nef* double mutant because depletion of mannose on Env did not rescue spread under conditions that were most comparable to our ten day spreading infections. The defect in spread is likely due to pleiotropic effects of Nef that disrupt interference by the HIV receptors, CD4, CXCR4 and CCR5 (*Lama et al., 1999*; *Michel et al., 2005*; *Venzke et al., 2006*) combined with the reduced infectivity of the mannose deficient Env.

Finally, we asked whether the mannose-deficient Env had increased stability in primary human MDM lacking Vpr and/or Nef by western blot analysis. We found that the Env mutant (N230D. N339E) was more stable in the absence of Vpr (*Figure 6I*, right side, black bars) and Nef (*Figure 6I*, right side, gray bars) once differences in infection frequency were accounted for by matching pr55 expression in the dilution series. These data provide strong support for a model in which MR restricts Env expression via direct interaction with high-mannose residues on Env and this restriction is counteracted by Vpr and Nef.

## Silencing MR alleviates restriction of Env in primary human MDM lacking Vpr

To directly test the hypothesis that MR is a restriction factor in MDM that is counteracted by Vpr, we examined the effect of MR silencing on Env expression in HIV-infected MDM lacking Vpr. Consistent with our hypothesis, we observed that silencing MR stabilized Env relative to Gag pr55 (*Figure 7A*). These results support the conclusion that the Env restriction observed in the absence of Vpr is dependent on expression of MR.

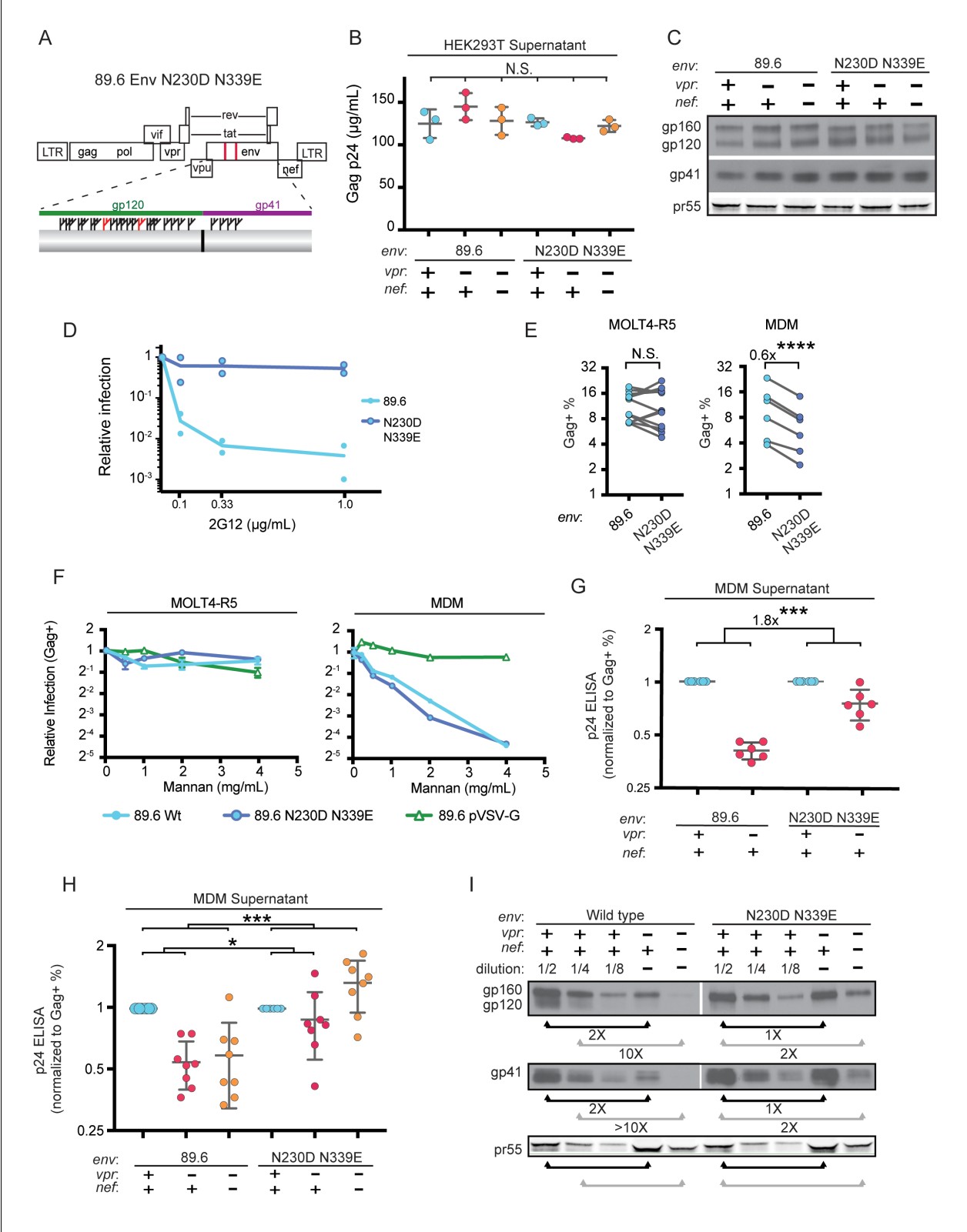

**Figure 6.** Deletion of N-linked glycosylation sites in *env* reduces the requirement for Vpr and Nef for virion release and Env expression in HIV-1 infected primary human MDM. (**A**) Upper panel, diagram of HIV genome encoding the mutations N230D and N339E (indicated in red) to prevent N-linked glycosylation at those sites. Lower panel, diagram of HIV 89.6 N230D N339E mutant Env protein. Branched symbols represent N-linked glycans. (**B**) Summary graph showing virion release from 293Ts transfected with the indicated HIV construct as measured by p24 ELISA. (*n* = 3 experimental

*Figure 6 continued on next page*

*Figure 6 continued*

replicates). Statistical significance was determined by one-way ANOVA. (C) Western blot analysis of 293T transfected as in B. (D) Summary graph showing relative infection frequency of MOLT4-R5 T cells by the indicated HIV following treatment as indicated with the neutralizing antibody 2G12. The percentage of infected cells was measured by intracellular Gag stain and normalized to the untreated condition for each virus. (n = 2 independent experiments, both are plotted) (E) Summary graphs of relative infection of the indicated cell type by mutant or parental wild-type HIV. The frequency of infected cells was measured flow cytometrically by intracellular Gag stain and normalized to the wild-type virus. (n = 5 experimental replicates for MOLT4-R5; n = 2 experimental replicates for MDM from four independent donors). (F) Summary graph depicting relative infection of the indicated cell type by each virus plus or minus increasing concentrations of mannan as indicated. The frequency of infected cells was measured by intracellular Gag stain and normalized to the uninhibited (0 mg/mL mannan) condition for each virus. 89.6 pVSV-G indicates 89.6 Δenv pseudotyped with VSV-G protein. (n = 2 independent donors for 89.6 wild-type and 89.6 Δenv pVSV-G; n = 1 donor for 89.6 env N230D N339E) (G) Summary graph of virion release from primary human MDM following 10 days of infection by the indicated HIV as measured by p24 ELISA. Virion release was normalized to the infection frequency assessed flow cytometrically by intracellular Gag stain. The result for each vpr-null mutant was normalized to the vpr-competent virus encoding the same env. (n = 6 independent donors) (H) Summary graph of virion release from primary human MDM following 10 days of infection by the indicated HIV as measured by p24 ELISA. Virion release was normalized to the infection frequency assessed flow cytometrically by intracellular Gag stain. For this single round infection assay, all viruses were pseudotyped with YU2 Env and viral spread was blocked 48 hr later by addition of AMD3100 and maraviroc. (n = 8 independent donors) The result for each vpr-null or vpr-nef-null mutant was normalized to the vpr- and nef-competent virus encoding the same env. (I) Western blot analysis of MDM infected as in G. The lysates from the vpr-competent and nef-competent infections were diluted to facilitate comparisons to vpr- and nef-null mutants. (n = 2 independent donors) For summary graphs, the mean +/- standard deviation is shown. In panels E, G and H statistical significance was determined by a two-tailed, paired t-test *p=0.01, **p<0.01, ***p<0.001. The online version of this article includes the following figure supplement(s) for figure 6:

**Figure supplement 1.** Raw p24 ELISA and intracellular Gag stain data following infection by 89.6 N230D N339E.

Previous work in our laboratory demonstrated that restriction of Env in primary human MDM disrupted formation of virological synapses and cell-to-cell spread of HIV from infected MDM to T cells (*Collins et al., 2015*). Expression of Vpr alleviated these effects, dramatically increasing viral transmission – especially under conditions of low initial inoculum of free virus. To expand on these findings, we measured Vpr-dependent HIV-1 spread from primary human MDM to autologous T cells, as diagrammed in *Figure 7—figure supplement 1A*. Co-cultured cells were stained for CD3 to distinguish T cells and CD14 to distinguish MDM as shown in *Figure 7—figure supplement 1B*, accounting for differences in autofluorescent background in the two cell types by using isotype controls (*Figure 7—figure supplement 1C*) We confirmed our prior finding that Vpr enhances HIV-1 89.6 spread from MDM to T cells (*Figure 7—figure supplement 1D*) and extended this finding to the transmitted/founder (T/F) clone REJO (*Figure 7—figure supplement 1E*). Consistent with our previous findings, we observed that a higher frequency of T cells became infected following co-culture with infected MDM as compared to incubation with high titer cell free virus [47-fold (89.6, p=0.0002) and 38-fold (REJO, p=0.048)].

To determine whether Vpr stimulated spread from macrophages to T cells by counteracting MR restriction, we measured spread to T cells from macrophages in which MR had been silenced as diagrammed in *Figure 7B*. Using the gating strategy shown in *Figure 7—figure supplement 1B*, infected MDM and infected T cells were identified by intracellular Gag stain (*Figure 7C*). We found that silencing MR reduced the difference between wild type and Vpr-null infected macrophage spread to T cells from 7-fold (p=0.003) to 2-fold (p=0.02) (*Figure 7D*). These results provide strong evidence that MR is the previously described but unidentified restriction factor in macrophages that reduces HIV spread from macrophages to T lymphocytes in the absence of Vpr.

## Discussion

We previously reported that Env and Env-containing virions are degraded in macrophage lysosomes in the absence of Vpr, impairing virion release, virological synapse formation, and spread of HIV to T cells (*Mashiba et al., 2014*; *Collins et al., 2015*). Moreover, this requirement for Vpr was conferred to heterokaryons comprised of macrophages and permissive cells, suggesting the existence of a previously unidentified host restriction factor that is counteracted by Vpr in macrophages (*Mashiba et al., 2014*). Results presented here clearly define mannose receptor (MR) as the HIV restriction factor counteracted by Vpr in macrophages to enhance viral dissemination. We provide strong evidence that Env mannosylation is required for restriction of Env and virion release in macrophages in the absence of Vpr, and that MR silencing relieves a requirement for Vpr to overcome this

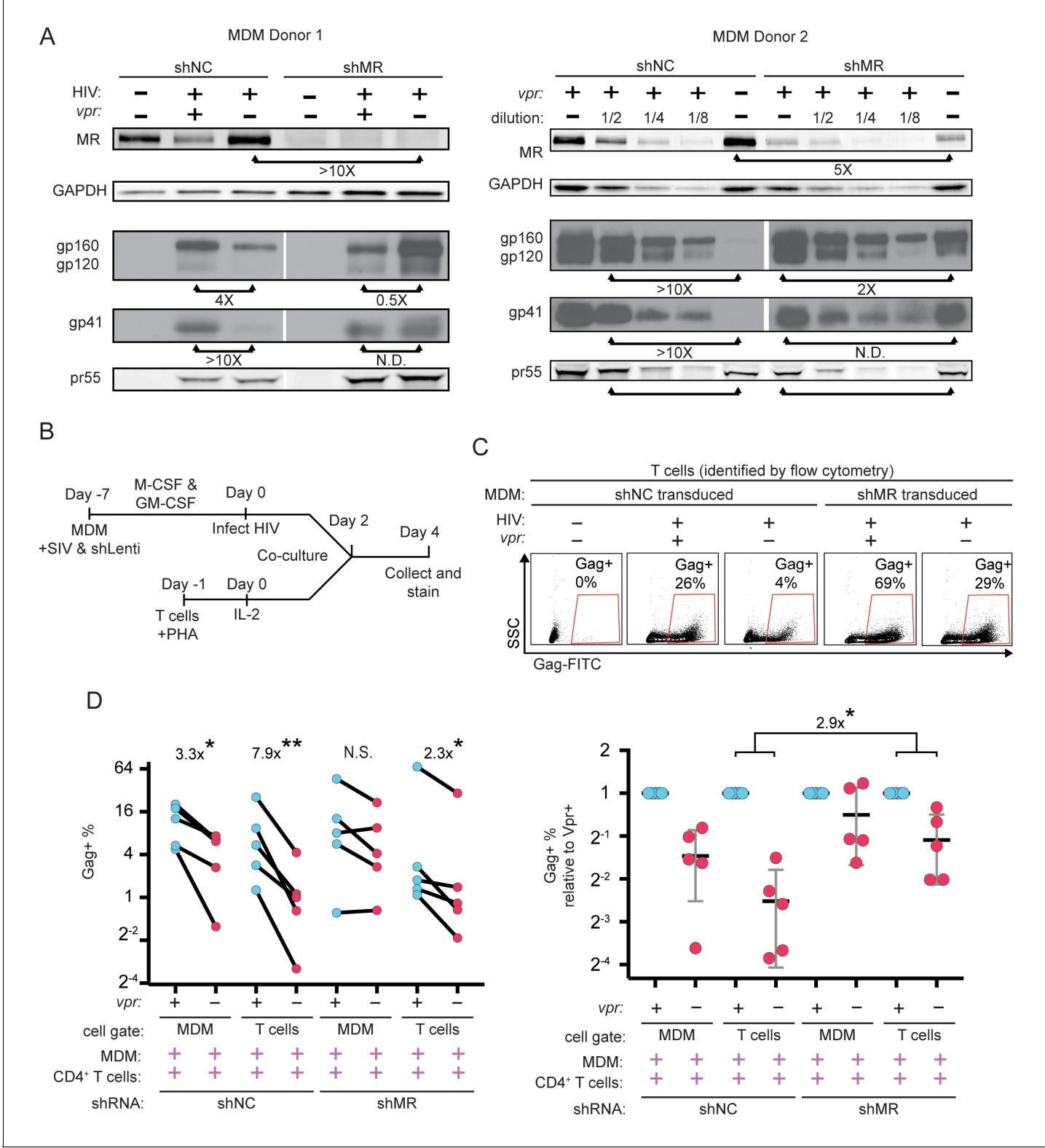

**Figure 7.** Knockdown of MR enhances Env expression and spread to T cells in *vpr*-null infection of MDM. (**A**) Western blot analysis of MDM from two independent donors treated with the indicated silencing vector and infected with the indicated HIV for 10 days. The shRNA sequences encoded by the negative control vector (shNC) and the MR silencing vector (shMR) are described in Methods. (**B**) Schematic diagram of experimental protocol used for silencing experiments. (**C**) Representative flow cytometric plots showing frequency of infected (Gag⁺) primary T cells following two days of co-culture with autologous, HIV 89.6 infected primary MDM. T cells were identified in co-culture by gating on CD3⁺ CD14⁻ cells as shown in *Figure 7—figure*

*Figure 7 continued on next page*

*Figure 7 continued*

*supplement 1B*. (D) Summary graph displaying relative infection of MDM and T cells as measured in C (*n* = 5 independent donors). Data in the left panel are unnormalized. In the right panel the data have been normalized to the wild-type condition for each donor and shRNA.

The online version of this article includes the following figure supplement(s) for figure 7:

**Figure supplement 1.** Cell-to-cell infection from macrophages to autologous CD4+ T cells is highly efficient and enhanced by Vpr.

restriction. Moreover, we confirm and extend a prior report that Nef also acts to downmodulate MR from the macrophage cell surface (*Vigerust et al., 2005*) and demonstrate that Vpr and Nef cooperate to counteract MR in an additive fashion through independent mechanisms.

Other investigators have reported that HIV inhibits *MRC1* transcription in macrophages and that MR inhibits virion egress upon exogenous expression in 293T cells (*Sukegawa et al., 2018*). In contrast to results we report here, the prior study observed effects on virions that were Env-independent and did not examine effects of Vpr on MR. In primary macrophages, Vpr-sensitive virion restriction only occurs when virions contain Env (*Mashiba et al., 2014*) and genetic changes in the *env* open reading frame – especially those that alter N-linked glycosylation sites – critically affect the requirement for Vpr. The effect of MR on Env and Env-containing virion release reported here helps explain previous observations that primate lentivirus infection reduces MR activity in humans (*Koziel et al., 1993*; *Koziel et al., 1998*) and monkeys (*Holder et al., 2014*). By confirming and extending our prior finding that Vpr-mediated stabilization of Env promotes macrophage to T cell spread (*Collins et al., 2015*) we also provide an explanation for how Vpr increases infection of human lymphoid tissue ex vivo (*Eckstein et al., 2001*; *Rucker et al., 2004*), which contain macrophages and T cells in a highly physiological, three-dimensional environment.

As Nef had already been shown to reduce MR surface expression (*Vigerust et al., 2005*), the observation that HIV encodes a second protein, Vpr, to reduce MR expression was unanticipated, but not unprecedented; other host proteins are known to be affected by more than one lentiviral accessory protein. The HIV receptor, CD4, is simultaneously targeted by Vpu, Nef and Env in HIV-1 (*Chen et al., 1996*) and tetherin is alternately targeted by Vpu, Nef, or Env in different strains of primate lentiviruses (*Harris et al., 2012*). Nef has also been shown to downmodulate the viral co-receptors CXCR4 (*Venzke et al., 2006*) and CCR5 (*Michel et al., 2005*), which may also interfere with Env expression and viral egress in infected cells. Nef's activity against CXCR4, CCR5, and MR presumably has the same ultimate purpose as its activity against CD4, namely to stabilize Env, enhance virion release and prevent superinfection of the producer cell (*Lama et al., 1999*; *Ross et al., 1999*). The impact of these deleterious interactions is clearly demonstrated by the profound loss of Env we observed in HIV-infected macrophages lacking both Vpr and Nef.

The need for both Vpr and Nef to counteract MR may be explained by the high level of MR expression, estimated at 100,000 copies per macrophage (*Stahl et al., 1980*). The potent combined effect likely derives from synergistic targeting of MR at two different stages of MR synthesis. Nef was shown to alter MR trafficking (*Vigerust et al., 2005*) and we show Vpr inhibits MR transcription.

In addition, our results suggest that maximal MR downmodulation is time-dependent in macrophages, which have the capacity to survive while infected for weeks; western blot analysis of whole cell lysates from saturated, ten-day infected cultures achieved a more striking reduction than was observed by flow cytometric analysis of five day cultures of macrophages infected with non-spreading viruses expressing GFP. This time dependency is potentially explainable in part by the very long half-life of MR [33 hr (*Lennartz et al., 1989*)] combined with the large amount of MR expressed per cell discussed above.

In sharp contrast to the effect we observed in MDM, Vpr did not affect MR protein levels when MR was expressed via a heterologous promoter in the 293T cell line, which is derived from human embryonic kidney cells and is not a natural target of HIV. The cell type selectivity in these experiments is likely due to differences in the promoters driving MR expression, however, we cannot rule out the existence of other macrophage specific pathways required to recreate the effect of Vpr on MR. Further work will be needed to examine these questions and determine other mechanistic details.

Our findings also implicate the Vpr binding protein VprBP/DCAF1 (*McCall et al., 2008*), a component of the cellular DCAF1-DDB1-CUL4 E3 ubiquitin ligase complex, in downmodulation of MR by

Vpr. This complex is required for most of the known functions of Vpr, including: disruption of the cell cycle, disruption of cellular DNA repair pathways in dividing cells (*Belzile et al., 2007*; *Hrecka et al., 2007*; *Le Rouzic et al., 2007*; *Wen et al., 2007*; *Lahouassa et al., 2016*; *Wu et al., 2016*; *Zhou et al., 2016*) and transcriptional inhibition of type I interferons in response to infection in macrophage cultures (*Laguette et al., 2014*; *Mashiba et al., 2014*). Additional research is now needed to determine how interactions between Vpr and DCAF1 mediate these pleiotropic effects.

Deleterious interactions between MR and Env that are alleviated by Vpr and Nef, likely occur along the secretory pathway and continue at the cell surface. This is based on previously published work showing that Env-containing virions are retained at the cell surface and targeted to lysosomes in macrophages lacking Vpr (*Collins et al., 2015*). Our prior studies also provided evidence that unprocessed Env gp160 is affected and targeted to lysosomal compartments albeit to a lesser degree (*Mashiba et al., 2014*). Because Env processing occurs via furin-mediated cleavage in the trans-Golgi network (TGN), the effect on unprocessed Env provides evidence that in addition to acting at the surface, MR likely also interacts with Env along the secretory pathway prior to its arrival and processing in the TGN.

MR's interaction with Env appears to be mediated by the unusually high density of N linked glycosylation sites on Env that retain high-mannose glycans, which is a known pathogen-associated molecular pattern (*Stahl and Ezekowitz, 1998*; *McGreal et al., 2006*). Here, we show that selective deletion of mannose residues alleviated the requirement for Vpr. Deletion of individual glycosylation sites is known to lead to changes in the processing of neighboring glycans and deletions at certain sites lead to larger than expected losses of oligomannose (*Balzarini, 2007*) presumably because their removal allows greater access to mannosidases and facilitates trimming of surrounding glycans. Selective pressure to maintain mannose residues on Env may be due to the enhanced attachment they mediate. Indeed, we provide strong evidence that Env's interaction with MR boosts initial infection of MDM. This finding is supported by a prior report that MR enhances HIV-1 binding to macrophages and transmission of the bound virus to co-cultured T cells (*Nguyen and Hildreth, 2003*). Our study adds to these findings by providing evidence that interactions with mannose binding receptors also enhance direct infection of macrophages. Moreover, the capacity of Vpr and Nef to mitigate the effect of detrimental intracellular interactions during viral egress limits the negative impact of retaining high-mannose on Env. In addition, the dense glycan packing, which is privileged from antibody recognition through immune tolerance, is believed to play a role in evasion of the antibody response (*Stewart-Jones et al., 2016*).

Because MR has both positive and negative effects on infection, the interpretation of some experiments examining spreading infection in the setting of MR silencing or mutations in Env that reduced mannose content were complex to interpret. Some donors had increased infection resulting from MR silencing whereas others had a small decrease at the ten-day time point (data not shown). By using viral systems that allowed us to focus independently on viral entry and exit, we nevertheless clearly discerned that MR can serve as a positive factor for entry and a negative factor for egress.

Thus far, all viral Envs we have tested (NL4-3, AD8 and 89.6) require Vpr for stable expression in macrophages except YU-2. We show here that genetically altering the mannose patch on 89.6 so that it mirrored changes in the YU-2 mannose patch altered the behavior of 89.6 to resemble that of YU-2 with respect to Vpr phenotypes. This is strong evidence supporting our model that Vpr alleviates deleterious interactions caused by the Env mannose patch. Interestingly, YU-2 was cloned from the central nervous system and 89.6 was directly cloned from peripheral blood. Because the blood-brain barrier limits exposure to antibodies, CNS isolates may have a diminished requirement for high mannose residues, which protect from antibody responses.

Here we also confirm and extend our prior observation (*Collins et al., 2015*) that co-culturing T cells with infected MDM boosted HIV infection compared to direct infection of T cells with cell-free virus. Similar to clone 89.6, T cell infection by the transmitter/founder virus REJO was enhanced by co-culture with MDM, and spread from MDM to T cells was enhanced by Vpr. In the context of natural person-to-person transmission, accelerated spread to T cells may be critical to establishing a persistent infection before innate and adaptive immune responses are activated. The strong selective pressure to retain Vpr despite its limited effect on T cell-only cultures indicates there is more to learn about the role of Vpr, macrophages and T/F viruses in HIV transmission and pathogenesis. Collectively, these studies suggest that novel therapeutic approaches to inhibit the activity of Vpr and Nef

in macrophages would potentially represent a new class of antiretroviral drug that could be an important part of a treatment or prophylactic cocktail.

# Materials and methods

### Key resources table

| Reagent type (species) | Designation | Source or reference | Identifiers | Additional Information |
|---|---|---|---|---|
| Recombinant DNA reagent | p89.6 | *Collman et al. (1992)* PMID: 1433527 | NIH AIDS Reagent Program 3552 | |
| Recombinant DNA reagent | p89.6 vpr-null | *Mashiba et al. (2014)*; PMID 25464830 | | |
| Recombinant DNA reagent | p89.6 nef-null | *Carter et al. (2010)*; PMID 20208541 | | |
| Recombinant DNA reagent | p89.6 vpr-nef-null | this paper | | Produces HIV 89.6 vpr-nef-null double mutant |
| Recombinant DNA reagent | p89.6 env N230D N339E | this paper | | Produces HIV 89.6 env N230D N339E mutant |
| Recombinant DNA reagent | p89.6 env N230D N339E vpr-null | this paper | | Produces HIV 89.6 env N230D N339E vpr-null mutant |
| Recombinant DNA reagent | p89.6 env N230D N339E vpr-nef-null | this paper | | Produces HIV 89.6 env N230D N339E vpr-nef-null mutant |
| Recombinant DNA reagent | pNL4-3 | *Adachi et al. (1986)*; PMID 3016298 | NIH AIDS Reagent Program 114 | |
| Recombinant DNA reagent | pNL4-3 env$^{YU2}$ | this paper | | Produces HIV NL4-3 env$^{YU2}$ chimera |
| Recombinant DNA reagent | pNL4-3 env$^{YU2}$ vpr-null | this paper | | Produces HIV NL4-3 env$^{YU2}$ vpr-null chimera |
| Recombinant DNA reagent | pHCMV-G | ATCC | 75497 | Expresses VSV-G |
| Recombinant DNA reagent | pCMV-HIV-1 | *Gasmi et al., 1999*; PMID 9971760 | | Expresses HIV structural proteins |
| Recombinant DNA reagent | pNL4-3 ΔGPE-GFP | *McNamara et al. (2012)*; PMID 22718820 | | |
| Recombinant DNA reagent | pNL4-3 ΔGPE-GFP vpr-null | this paper | | Produces NL4-3 ΔGPE vpr-null |
| Recombinant DNA reagent | pNL4-3 ΔGPE-GFP vpr-Q65R | this paper | | Produces NL4-3 ΔGPE vpr-Q65R |
| Recombinant DNA reagent | pNL4-3 ΔGPE-GFP nef-null | this paper | | Produces NL4-3 ΔGPE nef-null |
| Recombinant DNA reagent | pNL4-3 ΔGPE-GFP vpr-nef-null | this paper | | Produces NL4-3 ΔGPE vpr-nef-null |
| Recombinant DNA reagent | pYU2 | *Li et al. (1991)*; PMID 1830110 | NIH AIDS Reagent Program 1350 | |
| Recombinant DNA reagent | pYU2 vpr-null | this paper | | Produces YU-2 vpr-null |
| Recombinant DNA reagent | pREJO.c/2864 | *Ochsenbauer et al. (2012)*; PMID 22190722 | NIH AIDS Reagent Program 11746 | |

*Continued on next page*

*Continued*

| Reagent type (species) | Designation | Source or reference | Identifiers | Additional Information |
|---|---|---|---|---|
| Recombinant DNA reagent | pREJO.c/2864 vpr-null | this paper | | Produces REJO vpr-null |
| Recombinant DNA reagent | pSIV3+ | *Pertel et al. (2011)*; PMID 21696578 | | |
| Recombinant DNA reagent | pSIV3+ vpr-null | this paper | | Produces SIV3+ vpr-null |
| Recombinant DNA reagent | pSPAX2 | *Pertel et al. (2011)*; PMID 21696578 | | |
| Recombinant DNA reagent | pAPM-1221 | *Pertel et al. (2011)*; PMID 21696578 | | Silences luciferase mRNA |
| Recombinant DNA reagent | pAPM-MRC1-C | this paper | | Silences MR mRNA |
| Recombinant DNA reagent | pMD2.G | *Pertel et al. (2011)*; PMID 21696578 | | Expresses VSV-G |
| Recombinant DNA reagent | pYU2 env | *Sullivan et al. (1995)*; PMID 7769703 | | |
| Recombinant DNA reagent | pCDNA3.hMR | *Liu et al. (2004)*; PMID 15047828 | | Expresses MR |
| Recombinant DNA reagent | pPROA-3FLAG-UNG2-EYFP | *Akbari et al. (2010)*; PMID 20466601 | | |
| Recombinant DNA reagent | pMSCV IRES-GFP | *Van Parijs et al., 1999*; PMID 10514006 | | |
| Recombinant DNA reagent | pMSCV 3xFLAG UNG2 IRES-GFP | this paper | | Expresses 3x FLAG-tagged UNG2 |
| Recombinant DNA reagent | pUC19 | *Norrander et al. (1983)*; PMID 6323249 | | |
| Chemical compound, drug | Ficoll-Paque Plus | GE Healthcare | 17-1440-02 | |
| Chemical compound, drug | rhM-CSF | R and D Systems | 216-MC-025/CF | |
| Chemical compound, drug | rhGM-CSF | R&D Systems | 215 GM-050 | |
| Chemical compound, drug | IL-2 | R&D Systems | 202-IL-010 | |
| Chemical compound, drug | phytohaemagglutinin-L | Calbiochem | 431784 | |
| Chemical compound, drug | Enzyme-free cell dissociation buffer, HBSS-based | ThermoFisher | 13150016 | |
| Chemical compound, drug | Blue loading buffer | Cell Signaling Technology | 7722 | |
| Chemical compound, drug | AMD3100 | *Hendrix et al., 2000*; PMID 10817726 | NIH AIDS Reagent Program 8128 | |
| Chemical compound, drug | Maraviroc | *Emmelkamp and Rockstroh, 2007*; PMID 17933722 | NIH AIDS Reagent Program 11580 | |

*Continued on next page*

*Continued*

| Reagent type (species) | Designation | Source or reference | Identifiers | Additional Information |
|---|---|---|---|---|
| Chemical compound, drug | streptavidin-HRP | Fitzgerald | 65R-S104PHRP | |
| Chemical compound, drug | 3,3′,5,5′-tetramethylbenzidine | Sigma | T8665-IL | |
| Chemical compound, drug | Gag p24 standard | ViroGen | 00177 V | |
| Chemical compound, drug | Protein G Column | GE Healthcare | 45-000-054 | |
| Commercial assay, kit | Q5 site-directed mutagenesis kit | New England Biolabs | E0554S | |
| Commercial assay, kit | EasySep Human CD14 Positive Selection Kit II | Stemcell Technologies | 17858 | |
| Commercial assay, kit | CD8 Dynabeads | ThermoFisher | 11147D | |
| Commercial assay, kit | RNeasy micro RNA isolation kit | Qiagen | 74004 | |
| Commercial assay, kit | qScript cDNA Supermix | Quantabio | 95048 | |
| Commercial assay, kit | TaqMan Gene Expression Master Mix | ThermoFisher | 4369016 | |
| Commercial assay, kit | EZ-link Micro Sulfo-NHS-Biotinylation kit | ThermoFisher | PI-21925 | |
| Sequence-based reagent | 896 dNef-F | this paper | PCR primer | CACCATTATCGTTTCAGACCCT |
| Sequence-based reagent | 896 dNef-R | this paper | PCR primer | TCTCGAGTTTAAACTTATAGCAAAGCCCTTTCCA |
| Sequence-based reagent | NL43 vprQ65R-Forward | this paper | PCR primer | AGAATTCTGCGACAACTGCTG |
| Sequence-based reagent | NL43 vprQ65R-Reverse | this paper | PCR primer | TATTATGGCTTCCACTCC |
| Sequence-based reagent | 3xFLAG UNG2 F | this paper | PCR primer | CTAGCTCGAGACCATGGACTACAAAGACCATGAC |
| Sequence-based reagent | 3xFLAG UNG2 R | this paper | PCR primer | GTTAACTCACAGCTCCTTCCAGTCAATGGGCTT |
| Sequence-based reagent | GeneExpression assay for ACTB | ThermoFisher | Hs99999903 | |
| Sequence-based reagent | GeneExpression assay for MRC1 | ThermoFisher | Hs00267207 | |
| Sequence-based reagent | GeneExpression assay for POL2A | ThermoFisher | Hs02786624 | |
| Sequence-based reagent | GeneExpression assay for GAPDH | ThermoFisher | Hs00172187 | |
| Sequence-based reagent | APM-MRC1-C Forward oligo | Sigma | DNA oligo | TCGAGAAGGTATATTGCTGTTGACAGTGAGCGAGTAACTTGACTGATAATCAATTAGTGAAGCCACAGATGTAATTGATTATCAGTCAAGTTACTTGCCTACTGCCTCGG |

*Continued on next page*

*Continued*

| Reagent type (species) | Designation | Source or reference | Identifiers | Additional Information |
|---|---|---|---|---|
| Sequence-based reagent | APM-MRC1-C Reverse oligo | Sigma | DNA oligo | AATTCCGAGGCAGTAGGC AAGTAACTTGACTGATAA TCAATTACATCTGTGGCT TCACTAATTGATTATCAG TCAAGTTACTCGCTCACT GTCAACAGCAATATACCTTC |
| Biological sample (*Homo sapiens*) | Buffy coats/LeukoPaks | New York Blood Center | | Buffy coats made from whole blood |
| Biological sample (adenovirus) | Adeno-nef | *Leonard et al. (2011)*; PMID 21543478 | | |
| Cell line (*Homo sapiens*) | HEK293T | ATCC | CRL-3216 | |
| Cell line (*Mus musculus*) | anti-gp41 hybridoma CHESSIE-8 | *Abacioglu et al. (1994)*; PMID 8068416 | NIH AIDS Reagent Program 526 | Purified ab used for WB (2 µg/mL) |
| Cell line (*Mus musculus*) | anti-p24 hybridoma 183-H12-5C | NIH AIDS Reagent Program | 1513 | Purified ab used for ELISA (1 µg/mL) |
| Cell line (*Mus musculus*) | anti-p24 hybridoma 31-90-25 | ATCC (discontinued) | HB-9725 | Purified ab used for ELISA (0.5 µg/mL) |
| Antibody | anti-mannose receptor-PE (mouse monoclonal) | Becton Dickinson | clone 19.2 cat# 555954 | FC (1 µL per test) |
| Antibody | anti-Gag CA p24-PE (mouse monoclonal) | Beckman Coulter | clone KC57 cat# 6604667 | FC (0.25 µL per test) |
| Antibody | anti-Gag CA p24-FITC (mouse monoclonal) | Beckman Coulter | clone KC57 cat# 6604665 | FC (0.25 µL per test) |
| Antibody | anti-FLAG (mouse monoclonal) | Sigma | clone M2 cat# F3165 | FC (1 µL per test), WB (1:1000) |
| Antibody | anti-CD4-APC (mouse monoclonal) | ThermoFisher | clone OKT4 cat# 17-0048-42 | FC (1 µL per test) |
| Antibody | anti-CD3-PacBlue (mouse monoclonal) | BioLegend | clone OKT3 cat# 317313 | FC (1 µL per test) |
| Antibody | anti-CD14-APC (mouse monoclonal) | BioLegend | clone HCD14 cat# 325608 | FC (1 µL per test) |
| Antibody | anti-mannose receptor (rabbit polyclonal) | Abcam | ab64693 | WB (1:1000) |
| Antibody | anti-rabbit-AF647 (goat polyclonal) | ThermoFisher | A21244 | WB (1:4000) |
| Antibody | anti-GAPDH (mouse monoclonal) | Abnova | clone 3C2 cat# H00002597-M01 | WB (1:2000) |
| Antibody | anti-mouse IgG1-AF647 (goat polyclonal) | ThermoFisher | A21240 | FC (1 µL per test), WB (1:4000) |
| Antibody | HIV-Ig (human polyclonal) | *Cummins et al. (1991)*; PMID 1995097 | NIH AIDS Reagent Program 3957 | WB (1:2000) |
| Antibody | anti-human-AF647 (goat polyclonal) | ThermoFisher | A21445 | WB (1:4000) |
| Antibody | anti-gp120 (sheep polyclonal) | *Hatch et al., 1992*; PMID 1374448 | NIH AIDS Reagent Program 288 | WB (1:1000) |
| Antibody | anti-sheep-HRP (rabbit polyclonal) | Dako | P0163 | WB (1:20,000) |

*Continued on next page*

*Continued*

| Reagent type (species) | Designation | Source or reference | Identifiers | Additional Information |
|---|---|---|---|---|
| Antibody | anti-gp41 (human monoclonal) | *Zwick et al. (2001)*; PMID 11602729 | NIH AIDS Reagent Program 11557 | WB (1:1000) |
| Antibody | anti-human (goat polyclonal) | ThermoFisher | 62–8420 | WB (1:10,000) |
| Antibody | anti-Nef (rabbit polyclonal) | *Shugars et al. (1993)*; PMID 8043040 | NIH AIDS Reagent Program 2949 | WB (1:1000) |
| Antibody | anti-Vpr (rabbit polyclonal) | Dr. Jeffrey Kopp | NIH AIDS Reagent Program 11836 | WB (1:1000) |
| Antibody | anti-rabbit (goat polyclonal) | ThermoFisher | 65–6120 | WB (1:10,000) |
| Antibody | anti-GFP (chicken polyclonal) | Abcam | ab13970 | WB (1:1000) |
| Antibody | anti-chicken-HRP (goat polyclonal) | ThermoFisher | A16054 | WB (1:10,000) |
| Antibody | anti-STING (rabbit monoclonal) | Cell Signaling Technology | clone D2P2F cat# 13647 | WB (1:500) |
| Antibody | anti-GBP5 (goat polyclonal) | Dr. Frank Kicrhhoff | sc-160353 | WB (1:500) |
| Antibody | anti-IFITM3 (rabbit polyclonal) | Proteintech | 11714–1-AP | WB (1:1000) |
| Antibody | anti-Env 2G12 (human monoclonal) | *Buchacher et al. (1994)*; PMID 7520721 | NIH AIDS Reagent Program 1476 | neutralization (1 µg/mL) |
| Software, algorithm | FlowJo 10 | BD | 10.6.1 | |
| Software, algorithm | ABI Sequence Detection Software | ThermoFisher | 1.4 | |
| Software, algorithm | ImageQuant TL | GE | 8.2.0 | |
| Software, algorithm | Photoshop CC | Adobe | 20.0.6 | |
| Software, algorithm | shRNA retriever | http://katahdin.mssm.edu/siRNA/RNAi.cgi?type=shRNA | | |

## Viruses, viral vectors, and expression plasmids

The following molecular clones were obtained via the AIDS Reagent Program: p89.6 [cat# 3552 from Dr. Ronald G. Collman), pNL4-3 (cat# 114 from Dr. Malcolm Martin), pREJO.c/2864 (cat# 11746 from Dr. John Kappes and Dr. Christina Ochsenbauer) and pYU2 (cat# 1350 from Dr. Beatrice Hahn and Dr. George Shaw). *Vpr*-null versions of 89.6, NL4-3, and YU2 were created by cutting the AflII site within *vpr* and filling in with Klenow fragment. The *vpr*-null version of REJO was created by doing the same at the AvrII site. A *nef*-null version of 89.6 was created by deleting *nef* from its start codon to the XhoI site. To do this, a PCR amplicon was generated from the XhoI site in *env* to *env*'s stop codon. The 3' reverse primer added a XhoI site after the stop codon. The 89.6 genome and the amplicon were digested with XhoI and ligated together. (5' primer CACCATTATCGTTTCAGACCCT and 3' primer TCTCGAGTTTAAACTTATAGCAAAGCCCTTTCCA). The NL4-3 env^YU2 chimera consists of the pNL4-3 plasmid in which the fragment from the KpnI site in *env* to the BamHI site in *env* has been replaced with the equivalent fragment of pYU-2. Because the KpnI site is not unique within the plasmid, the fragment from the SalI site to BamHI site (which are unique) was cloned into pUC19, the change was made in *env*, and the fragment from SalI to BamHI was inserted back into pNL4-3. To generate p89.6 N230D N339E a synthetic DNA sequence (ThermoFisher, Waltham, Massachusetts) was purchased commercially. The synthetic gene contained the following nucleotide mutations, counting from the start of 89.6 *env*: 694 A > G, 701 C > A, 1018 A > G, 1020 T > A. This

sequence was substituted into p89.6 using the KpnI and BsaBI sites within *env*. pSIV3+, pSPAX2, pAPM-1221 and pMD2.G were obtained from Dr. Jeremy Luban (*Pertel et al., 2011*). pSIV3+ *vpr*-null was generated using a synthesized DNA sequence (ThermoFisher) containing a fragment of the SIV genome in which the Vpr start codon was converted to a stop codon (TAG). This was substituted into pSIV3+ using the sites BstBI and SapI. pYU2 env was obtained from Dr. Joseph Sodroski (*Sullivan et al., 1995*). Creation of pNL4-3 ΔGPE-GFP was described previously (*Zhang et al., 2004*; *McNamara et al., 2012*). Notably, the transcript containing the *gfp* gene retains the first 42 amino acids of *env*, including the signal peptide, which creates a fully fluorescent Env-GFP fusion protein. The *vpr*-Q65R mutant of NL4-3 ΔGPE-GFP was created using the Q5 site-directed mutagenesis kit from New England Biolabs (Ipswich, MA). The forward primer was AGAATTCTGCGACAACTGCTG and the reverse primer TATTATGGCTTCCACTCC. After synthesis by PCR, the entire provirus was confirmed by sequencing.

pCDNA.3.hMR was obtained from Dr. Johnny J. He (*Liu et al., 2004*). pPROA-3FLAG-UNG2-EYFP was obtained from Dr. Marit Otterlei (*Akbari et al., 2010*) and 3x FLAG tagged UNG2 was amplified using the 5' primer CTAGCTCGAGACCATGGACTACAAAGACCATGAC, which added an XhoI site, and the 3' primer GTTAACTCACAGCTCCTTCCAGTCAATGGGCTT, which added an HpaI site. The amplicon was cloned into the XhoI and HpaI sites of pMSCV IRES-GFP (*Van Parijs et al., 1999*) to generate pMSCV 3xFLAG UNG2 IRES-GFP.

## Primary MDM and T cell isolation and culture

Leukocytes isolated from anonymous donors by apheresis were obtained from the New York Blood Center Component Laboratory. The use of human blood from anonymous, de-identified donors was classified as non-human subject research in accordance with federal regulations and thus not subjected to formal IRB review. Peripheral blood mononuclear cells (PBMCs) were purified by Ficoll density gradient. CD14+ monocytes were positively selected using a CD14 sorting kit (cat# 17858, StemCell Technologies, Vancouver, Canada) following the manufacturer's instructions. Monocyte-derived macrophages (MDM) were obtained by culturing monocytes in R10 [RPMI-1640 with 10% certified endotoxin-low fetal bovine serum (Invitrogen, ThermoFisher), penicillin (100 Units/mL), streptomycin (100 µg/mL), L-glutamine (292 µg/mL), carrier-free M-CSF (50 ng/mL, R and D Systems, Minneapolis, Minnesota) and GM-CSF (50 ng/mL, R and D Systems)] for seven days. Monocytes were plated at $5 \times 10^5$ cells/well in a 24 well dish, except for those to be transduced with lentivirus and puromycin selected, which were plated at $1 \times 10^6$ cells/well.

CD4+ T lymphocytes were prepared from donor PBMCs as follows: anti-CD8 Dynabeads (cat# 11147D, ThermoFisher) were used to deplete CD8+ T lymphocytes and the remaining cells, which were mainly CD4+ lymphocytes, were maintained in R10 until the time of stimulation. Lymphocytes were stimulated with 5 µg/mL phytohemagglutinin (PHA-L, Calbiochem, Millipore Sigma, Burlington, Massachusetts) overnight before addition of 50 IU/mL recombinant human IL-2 (R and D Systems).

## Cell lines

The 293T cell line was obtained from ATCC and independently authenticated by STR profiling. It was maintained in DMEM medium (Gibco) supplemented with 100 U/mL penicillin, 100 µg/mL streptomycin, 2 mM glutamine (Pen-Strep-Glutamine, Invitrogen), 10% fetal bovine serum (Invitrogen), and 0.022% plasmocin (Invivogen). The MOLT-R5 cell line was obtained from the NIH AIDS Reagent Repository, which confirmed the lot is mycoplasma negative. It was maintained in RPMI-1640 medium (Gibco) supplemented with 100 U/mL penicillin, 100 µg/mL streptomycin, 2 mM glutamine (Pen-Strep-Glutamine, Invitrogen), 10% fetal bovine serum (Invitrogen), and 0.022% plasmocin (Invivogen).

## Silencing by shRNA

Sequences within *MRC1* suitable for shRNA-based targeting were identified using the program available at http://katahdin.mssm.edu/siRNA/RNAi.cgi?type=shRNA maintained by the laboratory of Dr. Ravi Sachidanandam. The sequence chosen, 5'-AGTAACTTGACTGATAATCAAT-3' was synthesized as part of larger DNA oligonucleotides with the sequences TCGAGAAGGTATATTGCTGTTGACAG TGAGCGAGTAACTTGACTGATAATCAATTAGTGAAGCCACAGATGTAATTGATTATCAGTCAAG TTACTTGCCTACTGCCTCGG (forward) and AATTCCGAGGCAGTAGGCAAGTAACTTGACTGATAA

TCAATTACATCTGTGGCTTCACTAATTGATTATCAGTCAAGTTACTCGCTCACTGTCAACAGCAATA TACCTTC (reverse). These oligos were annealed, which created overhangs identical to those produced by digestion with the enzymes EcoRI and XhoI. This double stranded DNA oligomer was inserted into the EcoRI and XhoI sites of pAPM-1221 to generate pAPM-MRC1-C.

Short hairpin RNA-mediated silencing was performed as previously described (*Pertel et al., 2011*; *Mashiba and Collins, 2013*; *Collins et al., 2015*). Briefly, we spinoculated freshly isolated primary monocytes with VSV-G-pseudotyped SIV3+ *vpr*-null at 2500 rpm for 2 hr with 4 µg/mL polybrene to allow Vpx-dependent degradation of SAMHD1. Cells were then incubated overnight in R10 with M-CSF (50 ng/mL) and GM-CSF (50 ng/mL) plus VSV-G-pseudotyped lentivirus containing an shRNA cassette targeting luciferase (pAPM-1221 or 'shNC') or MR (pAPM-MRC1-C or 'shMR'). The following day, media was removed and replaced with fresh R10 with M-CSF (50 ng/mL) and GM-CSF (50 ng/mL). Three days later 10 µg/mL puromycin was added and cells were cultured for three additional days prior to HIV-1 infection. shRNA target sequences used: *Luciferase*: 5'-TACAAACGCTC TCATCGACAAG-3', *MRC1:* 5'-ATTGATTATCAGTCAAGTTACT-3'.

## Virus production

Virus stocks were obtained by transfecting 293T cells (ATCC, Manassas, Virginia) with viral DNA and polyethylenimine (PEI). Cells were plated at $2.5 \times 10^6$ cells per 10 cm dish and incubated overnight. The following day 12 µg of total DNA was combined with 48 µg of PEI, mixed by vortexing, and added to each plate of cells. For NL4-3 ΔGPE-GFP, cells were transfected with 4 µg viral genome, 4 µg pCMV-HIV, and 4 µg pHCMV-V (VSV-G expression plasmid). For SIV3+ *vpr*-null the cells were transfected with 10.5 µg of viral genome and 1.5 µg pHCMV-V. For shLentivirus (shNC or shMR) cells were transfected with 6 µg pAPM-1221 or pAPM-MRC1-C, 4.5 µg pSPAX2, and 1.5 µg pMD2.G. Viral supernatant was collected 48 hr post-transfection and centrifuged at 1500 rpm (500 x *g*) 5 min to remove cellular debris. SIV3+ *vpr*-null was pelleted by centrifugation at 14,000 rpm (23,700 x *g*) for 4 hr at 4°C and resuspended at 10x concentration. Virus stocks were aliquoted and stored at −80°C.

## Co-transfections

Co-transfections of HIV and MR or UNG2 were performed in 293T cells. Cells were plated at $1.6 \times 10^5$ per well in a 12-well dish. The following day 10 ng of pcDNA.3.hMR or 10 ng of pMSCV 3xFLAG UNG2 IRES-GFP, 250 ng of NL4-3 ΔGPE-GFP, and 740 ng pUC19 plasmid was combined with 4 µg PEI, mixed by vortexing, and added to each well. 48 hr later, cells were lifted using enzyme free cell dissociation buffer (ThermoFisher, cat# 13150016) and analyzed by flow cytometry or lysed in 500 µL blue loading buffer (cat# 7722, Cell Signaling Technology, Danvers, Massachusetts) and analyzed by western blot.

## HIV infections of MDM

Prior to infection, 500 µL of medium was removed from each well and this 'conditioned' medium was saved to be replaced after the infection. MDM were infected by equal inocula of HIV as measured by Gag p24 mass in 500 µL of R10 for 6 hr at 37°C. After 6 hr, infection medium was removed and replaced with a 1:2 mixture of conditioned medium and fresh R10. Where indicated, HIV spread was blocked by AMD3100 (10 µg/mL, AIDS Reagent Program cat# 8128) and/or maraviroc (20 µM, AIDS Reagent Program cat# 11580) added 48 hr post-infection and replenished with each media change every three days.

## Spin transduction of MDM with NL4-3 ΔGPE-GFP

MDM were centrifuged at 2500 rpm (1049 x *g*) for 2 hr at 25°C with equal volume of NL4-3 ΔGPE-GFP or an isogenic mutant in 500 uL total medium. Following infection, medium was removed and replaced with a 1:2 mixture of conditioned medium and fresh R10.

## Adenoviral transduction of MDM

Adenovirus was prepared by the University of Michigan Vector Core, and the transduction of MDM was performed as previously described (*Leonard et al., 2011*) at an MOI of 1000 based on 293T cell infection estimations and the concentration of particles as assessed by $OD_{280}$.

## Infection of T cells

Activated T cells were infected by two methods as indicated. For direct infection, $5 \times 10^5$ cells were plated per well with 50 µg HIV p24 in 500 µL R10 +50IU/mL of IL-2 and incubated at 37°C for 48 hr. For co-culture with autologous, infected MDM medium was removed from MDM wells and $5 \times 10^5$ T cells were added in 1mL R10 + 50IU/mL of IL-2. All T cell infections were collected 48 hr post infection.

## Flow cytometry

Intracellular staining of cells using antibodies directed against HIV Gag p24, MR and FLAG-UNG2 was performed by permeabilizing paraformaldehyde-fixed cells with 0.1% Triton-X in PBS for 5 min, followed by incubation with antibody for 20 min at room temperature. For Gag and MR, PE-conjugated primary antibodies were used. For FLAG-UNG2, cells were stained with a PE-conjugated goat anti-mouse IgG1 secondary antibody for 20 min at room temperature. Surface staining for CD4, CD3 and CD14 was performed before fixation as described previously (Collins et al., 2015). Flow cytometric data was acquired using a FACSCanto instrument with FACSDiva collection software (BD, Franklin Lakes, New Jersey) or a FACScan (Cytek, BD) with FlowJo software (TreeStar, Ashland, Oregon) and analyzed using FlowJo software. Live NL4-3 ΔGPE-GFP transduced cells were sorted using a FACSAria III (BD) or MoFlo Astrios (Beckman Coulter) and gating on GFP$^+$ cells.

## Quantitative RT-PCR

MDM sorted as described above in 'Flow cytometry' were collected into tubes containing RLT buffer (Qiagen, Hilden, Germany) and RNA was isolated using RNeasy Kit (Qiagen) with on-column DNase I digestion. RNA was reverse transcribed using qScript cDNA SuperMix (Cat #95048, Quantabio, Beverly, Massachusetts). Quantitative PCR was performed using TaqMan Gene Expression MasterMix (ThermoFisher, cat# 4369016) on an Applied Biosystems 7300 Real-Time PCR System using TaqMan Gene Expression primers with FAM-MGB probe. The primer/probe sets for *ACTB* (Hs99999903), *MRC1* (Hs00267207), *POL2A* (Hs02786624), and *GAPDH* (Hs00172187) were purchased from ThermoFisher. Reactions were quantified using ABI Sequence Detection software compared to serial dilutions of cDNA from mock-treated cells. Measured values for all genes were normalized to measured values of *GAPDH* or *ACTB* as indicated.

## Immunoblot

MDM cultures were lysed in Blue Loading Buffer (cat# 7722, Cell Signaling Technology), sonicated with a Misonix sonicator (Qsonica, LLC., Newtown, Connecticut), boiled for 5 min at 95°C and clarified by centrifugation at 8000 RPM (7,000 x *g*) for 3 min. Lysates were analyzed by SDS-PAGE immunoblot. The proteins MR, GAPDH and pr55 were visualized using AlexFluor-647 conjugated secondary antibodies on a Typhoon FLA 9500 scanner (GE, Boston, Massachusetts) and quantified using ImageQL (GE). The proteins gp160, gp120, gp41, Nef, Vpr, GFP, Env-GFP, STING, GBP5, and IFITM3 were visualized using HRP-conjugated secondary antibodies on film. Immunoblot films were scanned and the mean intensity of each band, minus the background, was calculated using the histogram function of Photoshop CC (Adobe, San Jose, California).

## Virion quantitation

Supernatant containing viral particles was lysed in Triton X lysis buffer (0.05% Tween 20, 0.5% Triton X-100, 0.5% casein in PBS). Gag p24 antibody (clone 183-H12-5C, AIDS Reagent Program cat# 1519 from Dr. Bruce Cheseboro and Dr. Hardy Chen) was bound to Nunc MaxiSorp plates (ThermoFisher cat# 12-565-135) at 4°C overnight. Lysed samples were captured for 2 hr and then incubated with biotinylated antibody to Gag p24 (clone 31-90-25, ATCC cat# HB-9725) for 1 hr. Clone 31-90-25 was biotinylated with the EZ-Link Micro Sulfo-NHS-Biotinylation Kit (ThermoFisher cat# PI-21925). Clones 31-90-25 and 182-H12-5C were purified using Protein G columns (GE Healthcare, cat# 45-000-054) following the manufacturer's instructions. Samples were detected using streptavidin-HRP (Fitzgerald, Acton, Massachusetts) and 3,3',5,5'-tetramethylbenzidine substrate (Sigma cat# T8665-IL). CAp24 concentrations were measured by comparison to recombinant CAp24 standards (cat# 00177 V, ViroGen, Watertown, Massachusetts).

## Antibodies

Antibodies to CAp24 (clone KC57-PE cat# 6604667 and KC57-FITC cat# 6604665, Beckman Coulter, Brea, California), CD3 (clone OKT3-Pacific Blue, cat# 317313, BioLegend, San Diego, California), CD14 (clone HCD14-APC, cat# 325608, BioLegend), CD4 (clone OKT4, cat#17-0048-42, Invitrogen, ThermoScientific), FLAG (clone M2, cat#F3165, Sigma), and MR (clone 19.2-PE, cat# 555954, BD) were used for flow cytometry. Antibodies to the following proteins were used for immunoblot analysis: MR (cat# ab64693, Abcam, Cambridge, Massachusetts), GAPDH (clone 3C2, cat# H00002597-M01, Abnova, Taipei, Taiwan), Gag pr55 (HIV-Ig AIDS Reagent Program cat# 3957), Env gp160/120 (AIDS Reagent Program cat# 288 from Dr. Michael Phelan), 89.6 and YU-2 Env gp41 (clone z13e1, AIDS Reagent Program cat# 11557 from Dr. Michael Zwick), NL4-3 Env gp41 (clone CHESSIE-8, AIDS Reagent Program cat# 526 from Dr. George Lewis), Vpr (AIDS Reagent Program cat# 3951 from Dr. Jeffrey Kopp), GFP (cat# ab13970, Abcam), Nef (AIDS Reagent Program cat# 2949 from Dr. Ronald Swanstrom), FLAG (clone M2, cat# F3165, Sigma), STING (D2P2F, cat# 13647, Cell Signaling Technology), GBP5 (sc-160353, which was a generous gift from Dr. Frank Kirchhoff), and IFITM3 (cat# 11714–1-AP, Proteintech, Rosemont, IL). Neutralizing antibody 2G12 (AIDS Reagent Program cat# 1476 from Dr. Hermann Katinger) was used at a 1 µg/mL at the time of infection. Antibody clone CHESSIE-8 was purified using Protein G columns (GE Healthcare, cat# 45-000-054) following the manufacturer's instructions.

## Acknowledgements

This research was supported by the NIH (5T32GM008353-27 to JL, F31AI125090-01 to JL, R01AI046998 to KLC, R56AI130004 to KLC, R21AI32379 to KLC, T32GM007863 to MM, T32AI007413 to MM and T32AI007528 to DRC). We are grateful to the University of Michigan Vector Core and the NIH AIDS Reagent Program for reagents and to the University of Michigan Flow Cytometry Core for assistance with experiments. We thank Dr. Estelle Chiari for providing pMSCV 3xFLAG UNG2 IRES-GFP

## Additional information

### Funding

| Funder | Grant reference number | Author |
|---|---|---|
| National Institute of Allergy and Infectious Diseases | R21AI132379 | Jay Lubow<br>Maria C Virgilio<br>Brian G Peterson<br>Kathleen L Collins |
| National Institute of Allergy and Infectious Diseases | R56AI120954 | David R Collins<br>Michael Mashiba<br>Kathleen L Collins |
| National Institute of Allergy and Infectious Diseases | R56AI130004 | Jay Lubow<br>Maria C Virgilio<br>Brian G Peterson<br>Kathleen L Collins |
| National Institute of Allergy and Infectious Diseases | RO1AI51192 | David R Collins<br>Michael Mashiba<br>Kathleen L Collins |
| National Institute of General Medical Sciences | T32GM008353-28 | Jay Lubow |

The funders had no role in study design, data collection and interpretation, or the decision to submit the work for publication.

### Author contributions

Jay Lubow, Conceptualization, Resources, Formal analysis, Investigation, Methodology; Maria C Virgilio, Formal analysis, Investigation, Methodology; Madeline Merlino, Investigation, Visualization; David R Collins, Conceptualization, Resources, Formal analysis, Funding acquisition, Investigation;

Michael Mashiba, Conceptualization, Resources, Formal analysis, Funding acquisition; Brian G Peterson, Formal analysis, Investigation; Zana Lukic, Valeri Terry, Resources, Investigation; Mark M Painter, Francisco Gomez-Rivera, Gretchen Zimmerman, Investigation; Kathleen L Collins, Conceptualization, Supervision, Funding acquisition

**Author ORCIDs**
Jay Lubow (iD) https://orcid.org/0000-0002-7125-453X
Maria C Virgilio (iD) https://orcid.org/0000-0002-4940-188X
David R Collins (iD) https://orcid.org/0000-0001-7903-346X
Brian G Peterson (iD) http://orcid.org/0000-0001-6871-2336
Valeri Terry (iD) https://orcid.org/0000-0002-2499-3882
Gretchen Zimmerman (iD) https://orcid.org/0000-0002-6014-9101
Kathleen L Collins (iD) https://orcid.org/0000-0002-1712-5809

**Decision letter and Author response**
Decision letter https://doi.org/10.7554/eLife.51035.sa1
Author response https://doi.org/10.7554/eLife.51035.sa2

## Additional files

**Supplementary files**
• Supplementary file 1. Key resources table.

• Transparent reporting form

**Data availability**
The study did not generate large data sets. All necessary data are included in figures.

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
