## [Decision Letter]

**Acceptance summary:**

This paper ties together observations that HIV replicates to low levels in monocytes and that the accessory gene vpr plays a role in this lower replication with recent findings that vpr counteracts a macrophage specific factor that targets Env. As a result, expression of Vpr increases steady-state levels of Env in infected macrophages. The present study focused on the mannose receptor, which was previously shown to bind HIV Env and is highly expressed in macrophages. Here they show that vpr downregulates mannose receptor expression, thereby avoiding an innate response driven by this receptor and increasing Env expression and HIV replication. Given that macrophages are an important cell type for HIV replication in vivo, this study advances our understanding of how virus replication is regulated in this key host cell of HIV.

**Decision letter after peer review:**

Thank you for sending your article entitled "Mannose receptor is a restriction factor of HIV in macrophages and is counteracted by the accessory protein Vpr" for peer review at *eLife*. Your article is being evaluated by 3 peer reviewers, and the evaluation is being overseen by a Reviewing Editor and Satyajit Rath as the Senior Editor.

This work from Lubow et all seeks to build on previous publications from the lab and identify an antiviral restriction factor of HIV in macrophage cells that is overcome by the accessory gene Vpr. They identified Manose Receptor (MR) as a target of both Vpr and Nef. They focus on Vpr because this role of Nef has been described and Vpr appears to have a similar level of effect as Nef. The authors make a solid case that MR affects Env expression in the absence of Vpr and Nef and that MR is responsible for limiting virus spread from MDM to T cells in the absence of Vpr. The overall finding is interesting although the results are disjointed at times – changes of virus types that make it harder to tell what is driving differences and some key controls are needed. But overall, with proper controls, the study identifies an interest virus/host cell interaction that may help explain aspects HIV replication in macrophages

The reviewers were generally positive but had a fairly long list of major concerns that would need to be addressed. Some common and important themes of the reviewers include: Some controls are needed, including for the degradation assay such as including a Vpr mutant control and a substrate control. Controls, transparency of the data were common themes of the reviews that need to be addressed. A more balanced view of the role of Nef and Vpr in this activity given Nef effect is at least as potent as Vpr so if it is modest, then Vpr is also modest. Figure 7 should be removed as it is peripheral and based on limited data. Ideally, including studies with more relevant viruses would strengthen the paper and at the very least, describing this limitation in the studies and the fact that different viruses were used for different experiments, which could confound interpretation in the Discussion. Likewise, statements about the work explaining vpr conservation seem unfounded overall, and even more so given the viruses tested, so these reviewers' issues needed to be addressed. Statistical tests should be applied to support conclusions where appropriate. Also, many errors were evident in the figures and legends and these should be looked at carefully.

Reviewer #1:

This paper pulls together observations that HIV replicates to low levels in monocytes and that vpr plays a role in this lower replication with recent findings that vpr counteracts a macrophage specific factor that targets Env. The present study focused on the mannose receptor, which was previously shown to bind HIV Env and is highly expressed in macrophages. Here they show that vpr downregulates MR expression, thereby avoiding an innate response driven by this receptor and increasing HIV replication. The overall finding is interesting although the results are disjointed at times – changes of virus types that make it harder to tell what is driving differences, a study of T cell infection that seems peripheral, as discussed below.

1) The author suggests Figure 5E shows differences in infectivity of select mutants. That is not evident from the figure as differences are subtle and less than two fold and there are no significant differences indicated. If the differences are not statistically significant the authors should tone down these claims that MR is important for entry/replication throughout. This would include removing –the end of the fifth paragraph of the Discussion.

2) In the shMR cells, it would be nice to see a traditional virus replication study to get a better sense of the KO effects, rather than a single time point since this is an important part of the paper.

3) I don't understand how Figure 7 fits in this study. It does not examine the MR or vpr and it now shifts to the study of T/F viruses, which are not used elsewhere (which is too bad – see below). It seems like an ad on. Related to that, the second paragraph of the Discussion does not appear to be accurate as they did not demonstrate effects of vpr and MR on virus spread to T cells. If they want to do the experiments to show this, that would be a good addition and allow for some nice conclusions and strengthen the paper. Otherwise the T cell spread aspect and Figure 7 should be removed. And the last paragraph of the Discussion.

4) 89.6 is such a weird HIV variant, it would be nice to see this using a virus that is actually representative of circulating macrophage-tropic viruses. Likewise, for NL4.3 – very culture adapted. This may be especially relevant given YU2 behaves differently from these viruses, which may or may not be due to the mannose patch differences since there are many other glycans on env. Making vpr and nef mutants and even the mannose patch mutants in REJO or a similar stain and showing and testing them in wt versus shMR cells would help tie this story together. The changing of strains for different experiments makes interpretations more difficult and a clean set of key experiments with one macrophage tropic virus that is not highly adapted in culture would make this a stronger paper.

Reviewer #2:

This work from Lubow et all seeks to build on previous publications from the lab and identify an antiviral restriction factor of HIV in macrophage cells that is overcome by the accessory gene Vpr. They identified Manose Receptor (MR) as a target of both Vpr and Nef. While the data is overall convincing to support repression of MR by both Vpr and Nef, the authors chose to focus on Vpr and largely ignore Nef, even though it has a greater effect on MR levels. Moreover, the mechanistic details of how Vpr alters MR are very underdeveloped, as are their investigations into how MR inhibits Env and virion production. Finally, the overall flow of the paper was disjointed, and felt like two partially complete studies fused into one. In summary, while the data showing that both Vpr and Nef downregulate MR expression are convincing, the mechanistic insights into how and why this is achieved by two distinct lentiviral accessory proteins is lacking. This would help to make the paper more well-rounded and complete.

1) It is difficult to assess whether the effect of Vpr on MR/MRC1 is direct, or simply due to the many cellular effects of Vpr expression. Moreover, the authors do not sufficiently investigate how Vpr inhibits MR expression. For example:

a) In Figure 1, the authors screened "candidate Env binding proteins" but only show data for MR. It would help to show additional candidates to highlight the specificity of MR degradation.

b) In Figure 2, roughly 20-45% of GFP+ (HIV infected cells) have decreased MR despite being targeted by two viral proteins – why is this, and what does it suggest about MR downregulation?

c) In Figure 2 and Figure 2—figure supplement 1 the authors use an overexpression assay in 293T cells and qRT-PCR of MRC1 to suggest that MR is not degraded by Vpr, however no controls for their degradation assay are shown (such as UNG), no western blots to show expression of MR or Vpr are shown, and only MRC1 mRNA levels are looked at, which is concerning given the large effects of Vpr on cellular transcription.

2) Figures 2 and 3 clearly show that Nef functions slightly better than Vpr at inhibiting MR expression, yet the authors state that the effect of Nef is "modest" and "the effect of Nef alone was relatively small". Subsequently, Nef is largely ignored throughout the rest of the manuscript. It would be beneficial for the authors to clearly demonstrate why they think Vpr, and not Nef, is the primary driver of the observed phenotypes.

3) There are experimental details and controls missing throughout the manuscript. While some of these are minor on their own, given that much of the data is normalized, this lack of transparency and controls makes the data difficult to truly assess overall.

Some examples: a) "Relative p24 release" is shown for many of the later figures, when p24 or GFP is used in early figures. Why is p24/GFP not shown throughout?

b) Many of the figures rely on quantification of western blots; however, HRP was used for many panels and compared to AlexaFluor antibodies. HRP is not a quantitative reagent, and should not be used as such.

c) Loading controls are missing from some western blots (ex 3A, 3E, 5C, 5I, etc).

d) MDMs are differentiated using both M-CSF and GM-CSF; this is concerning as those cytokines lead to different macrophage phenotypes (such as highlighted here https://www.jimmunol.org/content/188/11/5752). Figure 1 shows MDMs more like GM-CSF MDMs, but Figure 6 indicates MDMs that sit somewhere in the middle. In addition, the MDMs seem to be CD3 positive (Supplementary Figure 2), which they should not be.

e) It is unclear why the authors use 10-day single round infections, which is unconventional. Moreover, the authors do not explain their methodology. For example, are the entry inhibitors replenished in the 8 days between their addition and experimental completion? It is highly doubtful the inhibitors can last for 8 days, so if they are not changed then the infection will not likely be single round.

f) Different y-axis scales are used even when discussing the same assay like p24 ELISA results (i.e. 4A vs. 4E.). Sometimes log 10, log 2, or linear…

g) Figure 7 is n=1 donor. Given the slight differences observed, given how variable different donor primary cells can be, and given the conclusions generated from this figure, this must be repeated with multiple donors.

4) The authors repeatedly use 293T cells as a control to show the observed effects are macrophage specific. However, 293Ts are highly transformed, rapidly cycling, lab adapted cells that do not make a good biological control cell when comparing to primary MDMs. THP-1 +/- PMA (to show cycling vs. non-cycling monocytic cell lines) or even primary T cells would be a better control to allow for biologically relevant conclusions.

5) I am wrestling with the biological significance of MR in HIV replication, spread, and evolution, especially given the statement that this work "provide(s) an explanation for the evolutionary conservation of Vpr"… The authors highlight the lack of manose patch naturally found in the macrophage-tropic YU-2 HIV isolate. If this is indeed a central target of MR and YU-2 has evolved without the manose patch in order to circumvent this restriction, then one would assume other macrophage-tropic viruses would also be selected to have lost similar glycan structures to increase viral fitness and avoid MR restriction. Have the authors looked for this? That said, if macrophage-tropic viruses are more fit by losing the manose patch to avoid MR, then why would two accessory genes (Vpr and Nef) be needed to deplete MR through two potentially independent mechanisms? Have the authors looked at Vpr and Nef from these macrophage-tropic viruses to see if they still inhibit MR expression? This might help explain the potential discrepancy.

Reviewer #3:

The authors previously reported that HIV-1 Vpr counteracts an unidentified macrophage-specific factor that targets Env and Env-containing virions for lysosomal degradation. Thus, expression of Vpr increases steady-state levels of Env in infected MDM. In the current study, the authors investigate the possible involvement of mannose receptor (MR). MR expression was previously shown to be downmodulated in HIV-infected MDM and HIV Tat and Nef were implicated in this phenomenon. Tat was proposed to cause transcriptional repression of the MR promoter whereas Nef was shown to cause cell-surface downmodulation of MR without effect on MR steady-state levels. Here the authors confirm the reported effect of Nef on MR surface expression. The authors did not specifically look at a possible inhibitory effect of Tat on MR transcription and, as far as I am concerned, their data do not categorically rule out a contribution of Tat. However, the data presented here are indeed more consistent with the authors' conclusion that it is Vpr that induces transcriptional repression of MR. Somewhat surprisingly, Vpr appeared to be sufficient to cause efficient inhibition of MR expression in MDM in a spreading infection experiment (Figure 1C), but had only a partial effect on MR surface expression in a single cycle infection study (Figure 2D) and required the additional presence of Nef for efficient downmodulation of MR. I am not quite sure what to make of that. Nevertheless, the authors go on to demonstrate that the previously reported effect of Vpr on Env expression involves MR and silencing of MR reduced the requirement for Vpr for efficient Env expression. Overall, this is an interesting study that is timely and provides novel insights into the interplay of Env, Vpr, Nef, and mannose receptor. The authors make a convincing case that MR affects Env expression in the absence of Vpr and Nef and that MR is responsible for limiting virus spread from MDM to T cells in the absence of Vpr. The manuscript is well written and the experiments are for the most part well-designed and executed. I do have a few issues as detailed below that require the authors attention.

1) Figure 1: Multiple studies have documented that replication of Vpr-deficient HIV-1 in macrophages is severely restricted when compared to wild type (wt) virus (e.g. Connor et al., 1995). I find it therefore astonishing that in panels C and E of Figure 1, there is only a minor difference in the pr55 levels of cells infected with wt and vpr-null virus. At 10 days post infection I would have expected a much larger difference in the Gag levels. In the figure legend and in the text the authors talk about "matched infection frequencies" but they don't explain how this was accomplished.

2) Also, I find it interesting that the authors go out of their way to document differences in Env levels by showing not only gp160 (the Env equivalent of pr55) but, in addition gp120 and gp41. Yet, they don't show p24 (CA). Showing the entire Gag blots (or at least inclusion of CA) might be informative since p24 antibodies typically have a higher affinity to p24 than to the pr55 precursor.

3) How does the strong reduction of MR in Figure 1C at the protein level by just Vpr fit with the data from Figure 2D where both Nef and Vpr were required for efficient cell-surface down-modulation of MR in a single round infection?

4) Figure 2D and G: How do the authors explain that only a subset of GFP+ cells downregulate MR? They skirt this issue both in the text and in the Discussion. In their Discussion, the authors make a statement concerning the importance of Env expression for Vpr-sensitive restriction in MDM. Since the construct used here is Env-defective, cells should be stained for intracellular p24 to see whether the Gag expression pattern matches that of GFP or MR (obviously it can't be both).

5) Figure 1—figure supplement 1: The authors should include a Q65R mutant of Vpr in the analysis to strengthen the argument that Vpr-mediated downmodulation of MR is DCAF1-independent.

6) Figure 3C/D: Donor 2: The% Gag-positive cells on day 10 is almost identical for wt and vpr-null samples. In contrast, virus output in panel D on day 10 is significantly lower for the vpr-null virus. Does that mean virus spreads mostly in a cell-to-cell manner in MDM? Does it correlate with low MR levels?

7) Nef does a lot of things but I am not aware of any literature reporting an effect of Nef on Env stability. What did I miss?

8) Figure 3E: Deletion of Nef alone reduces Env expression almost as much as deleting both Nef and Vpr. How can this be explained? Is it correlated with effect on MR expression? A MR western blot should be included here. But even if the effect of Nef deletion involves MR how can this work if Nef only affects MR surface expression and not total cellular pool of MR?

9) Figure 5I: The authors conclude from this experiment that MR restricts Env expression via direct interaction with high-mannose residues on Env. Where in the cell is this happening? What happens if cells are treated with Brefeldin A, which traps Env and MR in the ER? Will it exacerbate/alleviate the effect of MR on Env?

10). “MR is the restriction factor”.… – I would suggest softening this statement to "MR is a restriction factor".

11) Discussion: "In the primary macrophage system, Vpr-sensitive virion restriction depends entirely on an intact env open reading frame (Mashiba et al., 2014)". What are the implications of this statement for the data shown in Figure 2 where Env was replaced by GFP?

12) Discussion:" The magnitude of the effect and the fraction of cells affected increased when both proteins were expressed". What does the "fraction of cells" statement mean? Are the authors suggesting that this is an "either-or" phenomenon where Nef downmodulates MR in one fraction of the cells and Vpr does the same but in another fraction and only together do we see downmodulation in the two fractions combined? That would be fantastic from a mechanistical point of view and would explain the results from but is very unrealistic, I am afraid.

13) The fourth paragraph of the Discussion is somewhat trivial and can be deleted without loss to content and clarity of the paper. Instead, the authors should simply include the Q65R control as suggested for Figure 1—figure supplement 1 above. That's a quick and easy experiment to do.

---

## [Author Response]

Reviewer #1:This paper pulls together observations that HIV replicates to low levels in monocytes and that vpr plays a role in this lower replication with recent findings that vpr counteracts a macrophage specific factor that targets Env. The present study focused on the mannose receptor, which was previously shown to bind HIV Env and is highly expressed in macrophages. Here they show that vpr downregulates MR expression, thereby avoiding an innate response driven by this receptor and increasing HIV replication. The overall finding is interesting although the results are disjointed at times – changes of virus types that make it harder to tell what is driving differences, a study of T cell infection that seems peripheral, as discussed below.1) The author suggests Figure 5E shows differences in infectivity of select mutants. That is not evident from the figure as differences are subtle and less than two fold and there are no significant differences indicated. If the differences are not statistically significant the authors should tone down these claims that MR is important for entry/replication throughout. This would include removing –the end of the fifth paragraph of the Discussion.

In revised Figure 6E (was 5E in the first draft), data from MDM from 6 independent donors are compiled. They show that infection by the mannose-depleted Env mutant is consistently reduced about 40% compared to the wild-type Env and that these differences are statistically significant in MDM but not in MOLT4-R5 T cells, which lack MR. Additionally, our claim is supported by Figure 5F showing that mannan inhibits infection of MR-expressing macrophages, but not infection of the MOLT4-R5 T cell line. Moreover, we show that VSV-G pseudotyping HIV rescues this inhibition, which makes sense as VSV-G is not mannosylated.

2) In the shMR cells, it would be nice to see a traditional virus replication study to get a better sense of the KO effects, rather than a single time point since this is an important part of the paper.

As shown in Figure 6 and discussed above, MR has both positive and negative effects on infection; our data indicate it is a positive factor for entry and a negative factor for egress, which is somewhat analogous to CD4 and chemokine receptors in the sense that they are needed for entry but can interfere with viral egress and spread. Thus, the effects of silencing on overall infection are complicated to interpret. Some donors had increased infection resulting from MR silencing whereas others had a small decrease at the ten-day time point. Variations in baseline MR expression likely also contributed to a degree of heterogeneity in our results. We added a paragraph in the Discussion to address this interesting point.

3) I don't understand how Figure 7 fits in this study. It does not examine the MR or vpr and it now shifts to the study of T/F viruses, which are not used elsewhere (which is too bad – see below). It seems like an ad on. Related to that, the second paragraph of the Discussion does not appear to be accurate as they did not demonstrate effects of vpr and MR on virus spread to T cells. If they want to do the experiments to show this, that would be a good addition and allow for some nice conclusions and strengthen the paper. Otherwise the T cell spread aspect and Figure 7 should be removed. And the last paragraph of the Discussion.

In response to the reviewer’s concern, we added Vpr data and moved these results as they largely confirmed findings from a prior study (Collins et al., 2015). These data are now included in Figure 7—figure supplement 1. In addition, we added data from additional donors to revised Figure 7, which strengthened our conclusion that Vpr modulated spread from macrophages to T lymphocytes in an MR-dependent manner.

4) 89.6 is such a weird HIV variant, it would be nice to see this using a virus that is actually representative of circulating macrophage-tropic viruses. Likewise, for NL4.3 – very culture adapted. This may be especially relevant given YU2 behaves differently from these viruses, which may or may not be due to the mannose patch differences since there are many other glycans on env. Making vpr and nef mutants and even the mannose patch mutants in REJO or a similar stain and showing and testing them in wt versus shMR cells would help tie this story together. The changing of strains for different experiments makes interpretations more difficult and a clean set of key experiments with one macrophage tropic virus that is not highly adapted in culture would make this a stronger paper.

In previously published work, we demonstrated that 89.6, NL4-3 and AD8 behave similarly with respect to these assays. In other words, Vpr from each of these molecular clones similarly stabilizes Env, promotes virion release and stimulates spread from T cells to macrophages (Mashiba et al., 2014) and (Collins et al., 2015). In the revised manuscript, we provide new data on the transmitter/founder virus REJO that demonstrate it behaves similarly.

Thus far, YU2 is the only variant we have tested that has a diminished requirement for Vpr in macrophages and this property may be unique to this molecular clone. We show here that genetically altering the mannose patch on 89.6 so that it mirrored changes in the YU-2 mannose patch altered the behavior of 89.6 to resemble that of YU-2 with respect to Vpr phenotypes. This is strong evidence supporting our model that Vpr alleviates deleterious interactions caused by the Env mannose patch. Interestingly, YU-2 was cloned from the central nervous system (CNS) and 89.6 was directly cloned from peripheral blood. Because the blood-brain barrier limits exposure to antibodies, CNS isolates may have a diminished requirement for high mannose residues, which protect from antibody responses. We have revised the Discussion to include these points. In future experiments we will commit a focused effort towards examining a panel of CNS-derived HIV isolates to further test this interesting hypothesis.

Reviewer #2:This work from Lubow et all seeks to build on previous publications from the lab and identify an antiviral restriction factor of HIV in macrophage cells that is overcome by the accessory gene Vpr. They identified Manose Receptor (MR) as a target of both Vpr and Nef. While the data is overall convincing to support repression of MR by both Vpr and Nef, the authors chose to focus on Vpr and largely ignore Nef, even though it has a greater effect on MR levels. Moreover, the mechanistic details of how Vpr alters MR are very underdeveloped, as are their investigations into how MR inhibits Env and virion production. Finally, the overall flow of the paper was disjointed, and felt like two partially complete studies fused into one. In summary, while the data showing that both Vpr and Nef downregulate MR expression are convincing, the mechanistic insights into how and why this is achieved by two distinct lentiviral accessory proteins is lacking. This would help to make the paper more well-rounded and complete.1) It is difficult to assess whether the effect of Vpr on MR/MRC1 is direct, or simply due to the many cellular effects of Vpr expression. Moreover, the authors do not sufficiently investigate how Vpr inhibits MR expression. For example:a) In Figure 1, the authors screened "candidate Env binding proteins" but only show data for MR. It would help to show additional candidates to highlight the specificity of MR degradation.

In revised Figure 1—figure supplement 1, we provide these data, which provide evidence for a selective effect of Vpr on MR. We also provide additional RNA data for cellular genes other than MR that supports this conclusion (Figures 3 and Figure 3—figure supplement 1).

b) In Figure 2, roughly 20-45% of GFP+ (HIV infected cells) have decreased MR despite being targeted by two viral proteins – why is this, and what does it suggest about MR downregulation?

The reviewer is correct about these results collected 5 days post transduction. Western blot analysis of whole cell lysates from our ten-day culture (Figure 1D) suggests more uniform effects at longer time points and suggest MR downregulation is time-dependent in macrophages, which tolerate HIV infection for weeks. This time dependency is potentially explainable in part by very long half-life of MR (33 hours) (Lennartz, Cole and Stahl, 1989). Additionally, as we mention in the Discussion, MR is present at very high levels on macrophages (100,000 per cell) and it is likely that Vpr and Nef both require high level expression to achieve maximal effect.

c) In Figure 2 and Figure 2—figure supplement 1 the authors use an overexpression assay in 293T cells and qRT-PCR of MRC1 to suggest that MR is not degraded by Vpr, however no controls for their degradation assay are shown (such as UNG), no western blots to show expression of MR or Vpr are shown, and only MRC1 mRNA levels are looked at, which is concerning given the large effects of Vpr on cellular transcription.

An updated Figure 2—figure supplement 1 contains flow cytometry data and western blots demonstrating that under the conditions of our expression assay in 293T cells, Vpr degrades UNG2 but has no effect on MR.

As mentioned above, the qPCR data examining MR RNA expression in primary human macrophages have been strengthened significantly. We extended our analysis to additional cellular genes (GAPDH and POL2A) and increased the number of donors tested. We consistently observed that Vpr expression reduced MR mRNA levels but did not alter those of GAPDH or POL2A.

2) Figures 2 and 3 clearly show that Nef functions slightly better than Vpr at inhibiting MR expression, yet the authors state that the effect of Nef is "modest" and "the effect of Nef alone was relatively small". Subsequently, Nef is largely ignored throughout the rest of the manuscript. It would be beneficial for the authors to clearly demonstrate why they think Vpr, and not Nef, is the primary driver of the observed phenotypes.

Our conclusions are based on the compiled data, which summarizes results from multiple donors. Across 7 donors as shown in Figure 2E, the difference between Vpr and Nef-null viral mutants was not statistically significant. Our data indicate that both Vpr and Nef are required for MR downmodulation, but neither is sufficient. Compared to the Vpr-Nef-null double mutant, the Vpr-null mutant (which has Nef) had a small effect, as did the Nef-null mutant (which has Vpr). The results have been re-written to make this point clearer and the representative image in Figure 2D is more reflective of the compiled data.

While both Vpr and Nef play a role in MR downmodulation, the discovery of this role for Vpr has not been previously reported whereas Nef’s effects on MR are not novel. Thus, we focused on Vpr.

3) There are experimental details and controls missing throughout the manuscript. While some of these are minor on their own, given that much of the data is normalized, this lack of transparency and controls makes the data difficult to truly assess overall.Some examples: a) "Relative p24 release" is shown for many of the later figures, when p24 or GFP is used in early figures. Why is p24/GFP not shown throughout?

GFP is used in our single round infections shown in Figure 2 to identify the subset of cells that are infected by flow cytometry. Our full length, wild type viruses do not include any marker genes, so GFP cannot be used in any subsequent figures. In these cases, we use intracellular Gag in place of GFP to identify infected cells and this allows us to determine a rough assessment of “burst size” (p24 produced per infected cell frequency). In most figures the p24 production and infection rates varied significantly across donors, but p24 release per cell was always lower in the mutants than the wild type. To show this result clearly, all values for the mutants were normalized to Wt before being compiled. To improve transparency, the revised manuscript includes raw p24 ELISA and Gag+ frequency data. The corresponding data for Figures 5 and 6 are provided in Figure 5—figure supplement 1 and Figure 6—figure supplement 1 respectively. For Figure 4, these data were provided as part of the main figure.

b) Many of the figures rely on quantification of western blots; however, HRP was used for many panels and compared to AlexaFluor antibodies. HRP is not a quantitative reagent, and should not be used as such.

We agree that HRP and ECL have a lower dynamic range than fluorescent antibodies coupled with imaging systems. However, AlexaFluor secondary antibodies did not produce reliable results for the Env proteins. To overcome this limitation, we made great effort to demonstrate that we were operating under conditions of relative linearity by providing serial dilutions of wild-type infected lysates for blots stained with ECL. This semi-quantitative analysis provided reproducible fold-differences between wild type and mutant Env expression levels across multiple donors.

c) Loading controls are missing from some western blots (ex 3A, 3E, 5C, 5I, etc).

All western blots for host proteins have GAPDH as a loading control. For western blots of viral proteins, we feel that pr55 is the most useful control as it is a proxy for viral protein expression. For blots that analyze both host and viral proteins, GAPDH and pr55 are both shown.

d) MDMs are differentiated using both M-CSF and GM-CSF; this is concerning as those cytokines lead to different macrophage phenotypes (such as highlighted here https://www.jimmunol.org/content/188/11/5752). Figure 1 shows MDMs more like GM-CSF MDMs, but Figure 6 indicates MDMs that sit somewhere in the middle.

While differences were seen between macrophages treated with each cytokine individually in the published paper provided by the reviewer, in our study all donor macrophages were treated with both cytokines. Similar to other investigators, we have noted significant donor variability but we do not have any reason to believe this is due to the cytokines used to prepare them. The dual cytokine protocol we used for macrophage differentiation and infection was communicated to us by the Siliciano lab as an experimental approach that facilitates macrophage infection by HIV. This protocol is fairly standard in the field, being utilized and published by other HIV investigators e.g. (Lahouassa, Daddacha et al. 2012, Mashiba et al., 2014, Collins et al., 2015, Chougui, Munir-Matloob et al., 2018, Clayton, Collins et al. 2018).

In addition, the MDMs seem to be CD3 positive (Supplementary Figure 2), which they should not be.

Macrophages have more autofluorescence than T cells because of their larger size. We have added flow cytometry plots (Figure 7—figure supplement 1C) to demonstrate that the PacBlue signal observed in macrophages is roughly equal in unstained, isotype-control stained, and anti-CD3-stained MDM, indicating there is no significant CD3 expression on MDMs as expected. By contrast PBMCs show a clear CD3-specific signal compared to isotype control, indicating CD3 expression.

e) It is unclear why the authors use 10-day single round infections, which is unconventional. Moreover, the authors do not explain their methodology. For example, are the entry inhibitors replenished in the 8 days between their addition and experimental completion? It is highly doubtful the inhibitors can last for 8 days, so if they are not changed then the infection will not likely be single round.

We thank the reviewers for pointing out these methodological points that are potentially confusing to readers. We have clarified in the revised manuscript that this methodology was chosen to make the single round infection experiments as similar as possible to the 10 day spreading infections, which we believed showed stronger phenotypes due to the long half-life of mannose receptor. Also, the legends and Materials and Methods have been amended to clarify that entry inhibitors were replenished in culture medium every 3 days.

f) Different y-axis scales are used even when discussing the same assay like p24 ELISA results (i.e. 4A vs. 4E.). Sometimes log 10, log 2, or linear…

All ELISA results from MDM are now displayed on a log scale.

g) Figure 7 is n=1 donor. Given the slight differences observed, given how variable different donor primary cells can be, and given the conclusions generated from this figure, this must be repeated with multiple donors.

In response to this concern, we performed additional experiments to increase the number of donors included in this analysis. In the revised manuscript, these data were moved to Figure 7—figure supplement 1E because they largely replicate our previously published results with a different HIV strain.

4) The authors repeatedly use 293T cells as a control to show the observed effects are macrophage specific. However, 293Ts are highly transformed, rapidly cycling, lab adapted cells that do not make a good biological control cell when comparing to primary MDMs. THP-1 +/- PMA (to show cycling vs. non-cycling monocytic cell lines) or even primary T cells would be a better control to allow for biologically relevant conclusions.

The 293T cells were useful due to their high transfectability to demonstrate that the mutations we made in various genes did not disrupt expression of other HIV proteins unintentionally. We will clarify that as the primary conclusion of these experiments in the revised manuscript.

We have tested THP-1 cells; however, they were difficult to infect with non-pseudotyped wild type viruses and do not express MR or demonstrate Vpr dependent phenotypes. We do include infections of the T cell line MOLT4-R5 in Figure 5 and we use primary T cells as controls in Figure 7. In two prior publications we demonstrated that activated primary T cells do not display Vpr infection and virion production phenotypes when cultured alone under the conditions of our assay (Mashiba et al., 2014, Collins et al., 2015). We considered performing western blots using primary T cells to examine the effect of Vpr-Nef double mutants on Env. However, it’s known that Nef downmodulates CD4, which is expressed at high levels in T cells and this activity of Nef likely stabilizes Env expression in T cells. Thus, this experiment is of limited utility in the context of the question being asked here. A paragraph in the Discussion (3^rd^ paragraph) has been amended to discuss how effects of Nef on MR likely contribute to but do not fully explain the downmodulation of Env we observe in the Nef and Nef-Vpr double mutants. Beyond that, we respectfully request clarification about what question the reviewer wants addressed that affects our conclusion that Vpr overcomes a restriction of HIV infection in macrophages that is mediated by MR.

5) I am wrestling with the biological significance of MR in HIV replication, spread, and evolution, especially given the statement that this work "provide(s) an explanation for the evolutionary conservation of Vpr"… The authors highlight the lack of manose patch naturally found in the macrophage-tropic YU-2 HIV isolate. If this is indeed a central target of MR and YU-2 has evolved without the manose patch in order to circumvent this restriction, then one would assume other macrophage-tropic viruses would also be selected to have lost similar glycan structures to increase viral fitness and avoid MR restriction. Have the authors looked for this? That said, if macrophage-tropic viruses are more fit by losing the manose patch to avoid MR, then why would two accessory genes (Vpr and Nef) be needed to deplete MR through two potentially independent mechanisms? Have the authors looked at Vpr and Nef from these macrophage-tropic viruses to see if they still inhibit MR expression? This might help explain the potential discrepancy.

Thus far, all the viruses we have tested in this and previous publications have yielded a consistent pattern with respect to Env restriction in macrophages and Vpr alleviation of that restriction with the exception of YU-2.

As we have written in the revised manuscript, we hypothesize that YU-2 has distinct characteristics with respect to the mannose patch because of its localization within the CNS, which changes the balance of selective factors experienced by the virus. Macrophage-tropic HIVs are selected within the CNS because macrophages are abundant and T cells are rare within this compartment (Joseph, Arrildt et al. 2014). Highly macrophage tropic HIVs evolve the capacity to bind low levels of CD4, and YU2 is one example of this phenomenon (Thomas, Dunfee et al. 2007).

The mannose patch is beneficial because it is protective against antibody recognition. Thus, antibodies serve as a selective force for maintenance of the mannose patch. Because antibodies are expressed at very low levels in the CNS they likely impose substantially less selective pressure on HIVs present in the CNS, such as YU-2.

In addition, interactions between MR and Env are beneficial to the virus because these interactions facilitate initial infection by free virus. (Figures 6E and 6F), Because YU-2 utilizes low levels of CD4 so efficiently, YU-2 likely benefits less from interactions between MR and Env.

Through Nef and Vpr, the virus has evolved mechanisms to derive maximal benefit from the positive effects of the mannose patch, while avoiding detrimental ones. We clarified these points in the revised manuscript.

Reviewer #3:The authors previously reported that HIV-1 Vpr counteracts an unidentified macrophage-specific factor that targets Env and Env-containing virions for lysosomal degradation. Thus, expression of Vpr increases steady-state levels of Env in infected MDM. In the current study, the authors investigate the possible involvement of mannose receptor (MR). MR expression was previously shown to be downmodulated in HIV-infected MDM and HIV Tat and Nef were implicated in this phenomenon. Tat was proposed to cause transcriptional repression of the MR promoter whereas Nef was shown to cause cell-surface downmodulation of MR without effect on MR steady-state levels. Here the authors confirm the reported effect of Nef on MR surface expression. The authors did not specifically look at a possible inhibitory effect of Tat on MR transcription and, as far as I am concerned, their data do not categorically rule out a contribution of Tat. However, the data presented here are indeed more consistent with the authors' conclusion that it is Vpr that induces transcriptional repression of MR. Somewhat surprisingly, Vpr appeared to be sufficient to cause efficient inhibition of MR expression in MDM in a spreading infection experiment (Figure 1C), but had only a partial effect on MR surface expression in a single cycle infection study (Figure 2D) and required the additional presence of Nef for efficient downmodulation of MR. I am not quite sure what to make of that. Nevertheless, the authors go on to demonstrate that the previously reported effect of Vpr on Env expression involves MR and silencing of MR reduced the requirement for Vpr for efficient Env expression. Overall, this is an interesting study that is timely and provides novel insights into the interplay of Env, Vpr, Nef, and mannose receptor. The authors make a convincing case that MR affects Env expression in the absence of Vpr and Nef and that MR is responsible for limiting virus spread from MDM to T cells in the absence of Vpr. The manuscript is well written and the experiments are for the most part well-designed and executed. I do have a few issues as detailed below that require the authors attention.

We thank the reviewer for the positive comments. As discussed above, we provided clarification in the revised manuscript that both Nef and Vpr are required but neither are sufficient for maximal MR downregulation.

1) Figure 1: Multiple studies have documented that replication of Vpr-deficient HIV-1 in macrophages is severely restricted when compared to wild type (wt) virus (e.g. Connor et al., 1995). I find it therefore astonishing that in panels C and E of Figure 1, there is only a minor difference in the pr55 levels of cells infected with wt and vpr-null virus. At 10 days post infection I would have expected a much larger difference in the Gag levels. In the figure legend and in the text the authors talk about "matched infection frequencies" but they don't explain how this was accomplished.

Please see reply to reviewer 1’s comment #5. Briefly, as previously reported (Mashiba et al., 2014), the effect of Vpr on infection frequency is reduced over time as the virus saturates the macrophage culture. Additionally, we have observed that the requirement for Vpr can be minimized by increasing the viral MOI. Thus, by adjusting the inoculum and the duration of the infection, we can often achieve saturating infection of both viruses that allow more direct comparison than if the infection frequencies were very different. For more attenuated viruses, such as those lacking both Vpr and Nef or both Vpr and mutant Env, it was not possible to obtain comparable infection frequencies. These experiments required more adjustments to account for infection frequency differences. This was clarified in the revised manuscript.

2) Also, I find it interesting that the authors go out of their way to document differences in Env levels by showing not only gp160 (the Env equivalent of pr55) but, in addition gp120 and gp41. Yet, they don't show p24 (CA). Showing the entire Gag blots (or at least inclusion of CA) might be informative since p24 antibodies typically have a higher affinity to p24 than to the pr55 precursor.

Blots for Gag p24 were included in the analysis for a previously published paper by (Mashiba et al., 2014). In that study we demonstrated no consistent differences in intracellular p24 mediated by Vpr expression across multiple donors. However, we observed more variability with p24 western blots than pr55 western blots – perhaps because p24 is in the mature virion and can be variably retained in macrophages. Nevertheless, because of the variability, we utilize unprocessed pr55 to normalize for infection in our western blot analyses.

3) How does the strong reduction of MR in Figure 1C at the protein level by just Vpr fit with the data from Figure 2D where both Nef and Vpr were required for efficient cell-surface down-modulation of MR in a single round infection?

As mentioned above, Vpr and Nef are both required but neither is sufficient for maximal MR downregulation. The wild-type virus used in Figure 1C expresses both Nef and Vpr. The loss of Vpr alone causes a significant increase in MR expression relative to wildtype, which is consistent with our findings in Figure 2 that both Nef and Vpr are required to observe strong downmodulation of MR protein

4) Figure 2D and G: How do the authors explain that only a subset of GFP+ cells downregulate MR? They skirt this issue both in the text and in the Discussion. In their Discussion, the authors make a statement concerning the importance of Env expression for Vpr-sensitive restriction in MDM. Since the construct used here is Env-defective, cells should be stained for intracellular p24 to see whether the Gag expression pattern matches that of GFP or MR (obviously it can't be both).

Please see also responses to reviewer 2 question 2:

The assays depicted in Figure 2D and 2E (previously 2G) were performed 5 days post transduction whereas the western blots of whole cell lysates were from ten-day cultures (Figure 1D). These data indicate that more striking effects on MR are observed at later time points post infection and suggest MR downregulation is time-dependent in macrophages, which tolerate HIV infection for weeks. This time dependency is potentially explainable in part by the very long half-life of MR (33 hours) (Lennartz, Cole and Stahl, 1989). Additionally, as we mention in the Discussion, MR is present at very high levels on macrophages (100,000 per cell) and it is likely that Vpr and Nef both must achieve high level expression to achieve maximal effect.

Adding a gag stain to the flow cytometry depicted in Figure 2D and E would not be informative, because, as shown in 2A, this construct lacks Gag. GFP, which is in the env ORF as a fusion with the first few amino acids of Env, should effectively mark all infected cells.

5) Figure 1—figure supplement 1: The authors should include a Q65R mutant of Vpr in the analysis to strengthen the argument that Vpr-mediated downmodulation of MR is DCAF1-independent.

We have completed the experiment with the Q65R mutant as suggested. As shown in Figure 2D and E, the vpr-Q65R mutant has a similar defect to vpr-null, which is consistent with a model in which DCAF1 binding is required to downmodulate MR. This was the expected result given our previous findings that the vpr-Q65R mutation was defective at reversing the macrophage-selective restriction of Env expression (Mashiba et al., 2014).

6) Figure 3C/D: Donor 2: The% Gag-positive cells on day 10 is almost identical for wt and vpr-null samples. In contrast, virus output in panel D on day 10 is significantly lower for the vpr-null virus. Does that mean virus spreads mostly in a cell-to-cell manner in MDM? Does it correlate with low MR levels?

We have clarified this important point in the revised manuscript. At earlier time points, the infection rate in the Vpr mutant is lower, indicating that the lower p24 release correlates with slower spread. As discussed above, by day 10 we often see that Wt and Vpr-null infections approach parity because nearly all the cells in the culture become infected.

7) Nef does a lot of things but I am not aware of any literature reporting an effect of Nef on Env stability. What did I miss?

We agree that our results showing that mutating Nef reduces Env stability in macrophages are novel. A likely related effect of Nef in other cell types was previously reported (Lama, Mangasarian and Trono, 1999). This paper demonstrates that CD4 reduces Env levels in virions. This effect of CD4 is thought to be due to CD4 binding Env within infected cells. It is reversed by Nef and Vpu-mediated CD4 downmodulation. We discuss this and how it parallels our model that Env binding to MR interferes with Env trafficking in the revised manuscript. We previously demonstrated that without Vpr, Env and Env containing virions are trafficked to lysosomes and degraded at an accelerated rate. The Vpr/Nef-mediated downmodulation of MR reverses this effect.

8) Figure 3E: Deletion of Nef alone reduces Env expression almost as much as deleting both Nef and Vpr. How can this be explained? Is it correlated with effect on MR expression? A MR western blot should be included here. But even if the effect of Nef deletion involves MR how can this work if Nef only affects MR surface expression and not total cellular pool of MR?

Our data indicate that both Vpr and Nef are required for maximal MR downmodulation, but neither is sufficient. In the revised version of Figure 4E (previously 3E) we provide additional data from a second donor in which the effects of Nef and Vpr mutation are more equal. Similarly, in the single round experiments depicted in 2E we observe some variation in the relative activity of Nef and Vpr. Overall, the compiled results did not show a significant difference between the Vpr mutant and the Nef mutant.

Because the infection frequencies for the mutant viruses shown in this figure are quite low, adding western blots of MR to Figure 4E would not allow us to properly assess the effect of viral proteins on MR. The Nef-null single mutant is defective at spread due to the multifactorial effects of Nef. In the case of Nef-null only about half of MDMs are infected. In the case of Vpr-Nef-null <20% are infected. Thus, the whole cell lysates largely reflect MR levels of uninfected cells in those samples.

9) Figure 5I: The authors conclude from this experiment that MR restricts Env expression via direct interaction with high-mannose residues on Env. Where in the cell is this happening? What happens if cells are treated with Brefeldin A, which traps Env and MR in the ER? Will it exacerbate/alleviate the effect of MR on Env?

This is an interesting idea but given the broad effects of brefeldin on trafficking and the potential for complex indirect effects, results will be difficult to interpret. For example, brefeldin treatment of Nef expressing cells leads to non-specific reduction of most proteins found on the cell surface and a modest increase in MHC-I. The experiment does not answer the question as to where in the cell Nef is acting. This question asked by the reviewer is better addressed by considering the forms of Env that are affected as well as effects on Env-containing virions that we previously published. We previously showed that Env-containing virions are retained at the cell surface and targeted to lysosomes in macrophages lacking Vpr (Collins et al., 2015). Our prior studies also provided evidence that unprocessed Env is affected and targeted to lysosomal compartments albeit to a lesser degree (Mashiba et al., 2014). Because Env processing occurs via furin-mediated cleavage in the TGN, the effect on unprocessed Env provides evidence that MR interacts with Env along the secretory pathway prior to its arrival and processing in the TGN. A paragraph discussing this interesting question has been added to the revised manuscript.

10) “MR is the restriction factor” – I would suggest softening this statement to "MR is a restriction factor".

This line was softened as suggested.

11) Discussion: "In the primary macrophage system, Vpr-sensitive virion restriction depends entirely on an intact env open reading frame (Mashiba et al., 2014)". What are the implications of this statement for the data shown in Figure 2 where Env was replaced by GFP?

We have clarified in the revised manuscript that Env is required for us to observe effects of Vpr on virion release. We propose that this is because MR requires interactions with Env to retain virions. In our model, Env expression is not required for us to observe effects of Vpr or Nef on MR, as demonstrated in Figure 2E.

12) Discussion:" The magnitude of the effect and the fraction of cells affected increased when both proteins were expressed". What does the "fraction of cells" statement mean? Are the authors suggesting that this is an "either-or" phenomenon where Nef downmodulates MR in one fraction of the cells and Vpr does the same but in another fraction and only together do we see downmodulation in the two fractions combined? That would be fantastic from a mechanistical point of view and would explain the results from but is very unrealistic, I am afraid.

We have clarified our interpretation that Vpr and Nef act synergistically in the same cells. Our evidence suggests Vpr and Nef each has a partial effect on its own that is dramatically enhanced in the presence of the other factor. Regarding the fact that only a subset of GFP+ cells are dramatically affected on day five post-infection, as described above, this is likely related to the time-dependent nature of the process.

13) The fourth paragraph of the Discussion is somewhat trivial and can be deleted without loss to content and clarity of the paper. Instead, the authors should simply include the Q65R control as suggested for Supplementary Figure 1 above. That's a quick and easy experiment to do.

In response to the reviewer’s concern, we tested the ability of the Q65R mutant to downmodulate MR and determined that it behaves similarly to Vpr-null viruses which is consistent with our model (Figure 2E).